

**The impact threshold of the aerosol radiation forcing on the boundary layer**
**structure in the pollution region**
Dandan Zhao[†1,2]; Jinyuan Xin[*†1,2]; Chongshui Gong[3]; Jiannong Quan[4]; Yuesi Wang[1,2]; Guiqian
Tang[1], Yongxiang Ma[1], Lindong Dai[1], Xiaoyan Wu[1], Guangjing Liu[1], Yongjing Ma[1]
1 State Key Laboratory of Atmospheric Boundary Layer Physics and Atmospheric Chemistry (LAPC), Institute of
Atmospheric Physics, Chinese Academy of Sciences, Beijing 100029, China
2 University of Chinese Academy of Sciences, Beijing 100049, China
3 Institute of Arid Meteorology, China Meteorological Administration, Lanzhou 730020, China
4 Institute of Urban Meteorology, Chinese Meteorological Administration, Beijing, China
(†) These authors contributed equally to this study.
(*) Correspondence: Jinyuan Xin; email: xjy@mail.iap.ac.cn; phone: (+86)15810006545; address: #40 Huayanli,
Chaoyang District, Beijing 100029, China
**Abstract:** Recently, there has been increasing interest in the relation between
particulate matter (PM) pollution and atmospheric boundary layer (ABL) structure.
However, this has yet to be fully understood because most studies have been superficial.
This study aimed to qualitatively assess the interaction between PM and ABL structure
in essence, and to further quantitatively estimate the effects of aerosol radiative forcing
(ARF) on the ABL structure. Multi-episode contrastive analysis stated the key to
determining whether haze outbreak or dissipation was the ABL structure (i.e., stability
and turbulence kinetic energy (TKE)) satisfied relevant conditions. However, it seemed
that the ABL structure change was in turn highly related to the PM level and ARF. |SFC-
ATM| (SFC and ATM is respectively the ARF at the surface and interior of the
atmospheric column) is the absolute difference between ground and atmosphere layer
ARFs, and the change in |SFC-ATM| is linearly related to the PM mass concentration.
However, the influence of ARF on the boundary layer structure is nonlinear. With
increasing |SFC-ATM|, the TKE level exponentially decreased, which was notable in
the lower layers/ABL but disappeared above the ABL. Moreover, the threshold of the
ARF effects on the ABL structure was determined for the first time, namely, once |SFC-
ATM| exceeded $\sim$ 55 W m$^{-2}$, the ABL structure would quickly stabilize and would



thereafter change little with increasing ARF. The threshold of the ARF effects on the
boundary layer structure could provide useful information for relevant atmospheric
environment improvement measures and policies, such as formulating the objectives of
phased air pollution control.
**Keywords:** boundary layer structure; aerosol radiative forcing; threshold; haze
pollution
**1 Introduction**
Most areas in China, such as the North China Plain (Li et al., 2020; Xu et al., 2019),
have suffered from a poor air quality as a result of the rapid economic growth. Beijing,
as the Chinese capital and major city in the North China Plain, has frequently
experienced severe and persistent haze events, characterized by an exceedingly high
particulate matter (PM) mass loading suspended in near-surface air (Wang et al., 2018;
Zhong et al., 2018). As previous studies have found, air pollution episodes are the result
of secondary aerosol formation and adverse meteorological conditions (An et al., 2019;
Guo et al., 2014; Li et al., 2017; Wang et al., 2014; Zheng et al., 2015; Wang et al.,
2012). PM is concentrated in the atmospheric boundary layer (ABL) (Petaja et al., 2016;
Tie et al., 2017), which is the lower part of the troposphere and is directly affected by
the surface (Quan et al., 2013). The diffusion, transmission, and accumulation of
pollutants are closely linked to ABL structure (meteorological conditions) variation
(Han et al., 2009; Kotthaus and Grimmond, 2018; Zheng et al., 2017). Numerous
studies have revealed that the meteorological factors in the boundary layer influence
the formation of air pollution periods (Hua et al., 2016; Liu et al., 2016; Miao et al.,
2018; Wang et al., 2012; Wang et al., 2014; Zhang et al., 2018). For instance, the
aerosols concentrated in the ABL exhibit a strong negative relationship with the ABL
height (ABLH) that determines the volume available for pollutant dispersion (Haman
et al., 2014; Schaefer et al., 2009; Su et al., 2018; Tang et al., 2016). Heavy air pollution
episodes have always occurred with persistent temperature inversions (Xu et al., 2019;
Zhong et al., 2017). Weak/calm winds are important in the long-term increase in air
pollutants (Niu et al., 2010; Yang et al., 2016). Additionally, previous studies have



reported that severe air pollution is always highly related to a high atmospheric
humidity, which is one of the manifestations of stagnant ABL conditions (Tie et al.,
2017; Petaja et al., 2016). Moreover, the feedback/interaction mechanism between the
boundary layer structure and aerosol loading during severe pollution events has been
analyzed in previous studies (Huang et al., 2018; Liu et al., 2018; Zhong et al., 2018;
Zhao et al., 2019).
However, most of the work was performed through relationship analysis of the PM
concentration and meteorological factors and only considered certain pollution
processes. Few attempts have been made to examine the interaction between the ABL
and air pollution in terms of essential aspects. Since the ABL is directly influenced by
the surface, it is the only atmosphere layer characterized by turbulent activities, while
higher atmosphere layers are weakly turbulent because of the strongly stable
stratification (Munro, 2005). Thus, the ABL acts as a notable turbulence buffer coupling
the surface with the free atmosphere, and PM and gas pollutants are only suspended in
the ABL and are convectively spread throughout it. The evolution of the ABL structure,
which plays a key role in pollutant accumulation/diffusion, is substantially the change
in turbulent kinetic energy (TKE) in the ABL (Garratt et al., 1992). Therefore, we
systematically analyzed the way the ABL interacts with pollutants via contrastive
analysis of multiple haze episodes based on not only specific meteorological factors but
also turbulent activity profiles and atmospheric stability indicators. Moreover, the
change in solar radiation reaching the ground drives the diurnal ABL evolution
considering the variation in atmospheric stability (Andrews, 2000). Since a strong
aerosol radiative effect occurs on severe air pollution, the diurnal evolution of the
atmospheric thermodynamic status is greatly affected (Dickerson et al., 1997; Stone et
al., 2008; Wilcox et al., 2016). As previous studies have reported, the aerosol radiative
forcing (ARF) is also a critical parameter that can further modify the boundary layer
structure during haze episodes (Huang et al., 2018; Liu et al., 2018; Zhong et al., 2018).
However, the influence degree of the aerosol radiative effect on the boundary layer
structure remains unclear. Quantitatively determining the effects of ARF on the ABL



structure is urgently needed. Furthermore, this paper would analyze the interaction
between the ABL structure and air pollution using high-resolution and real-observation
datasets, such as temperature and humidity profiles of microwave radiometers,
horizontal and vertical wind vector profiles of Doppler wind lidar, ABL heights (ABLH)
and aerosol backscattering coefficient profiles of ceilometers. Wind profile lidar and
microwave radiometers have the advantage of providing direct and continuous
observations of the boundary layer over long periods of time and can characterize the
ABL structure up to 2-3 km (Pichugina et al., 2019; Zhao et al., 2019), compensating
for the deficiencies of previous research.
**2 Data and methods**
We conducted a two-month measurement campaign of the PM concentration and
aerosol optical depth (AOD) and obtained vertical profiles of atmospheric parameters
such as the temperature, humidity, wind vectors, atmospheric stability and TKE to
better understand how the boundary layer structure responds to aerosol radiative effects.
Figure S1 shows the observation site of the Tower Branch of the Institute of
Atmospheric Physics (IAP), Chinese Academy of Sciences (39°58′ N, 116°22′ E;
altitude: 58 m) and the sampling instruments in this study. The IAP site represents a
typical urban Beijing site and all the sampling instruments are placed at the same
location, and simultaneous monitoring is conducted. The algorithm of SBDART (Santa
Barbara DISORT Atmospheric Radiative Transfer) (Levy et al., 2007) is the core model
to calculate the radiative forcing parameters. Standard mid-latitude atmosphere is used
in SBDART in Beijing. AOD and Angstrom Exponent (AE) at 550 nm were obtained
from sun-photometer. Multiple sets of Single Scattering Albedo (SSA) and
backscattering coefficient were calculated based on MIE theory and surface albedo &
path radiation were read from MODIS (MOD04) which is used to calculate radiative
forcing at top of atmosphere (TOA). The TOA results were combined with MODIS
observations, the result which has the lowest deviation are defined as the actual
parameters of aerosols and this set of parameters would be used to calculate the
radiative forcing at the surface, top and interior of the atmospheric column (Gong et al.,


2014). Hourly radiative forcing parameters, including the ARF at the top (TOA),
surface (SFC) and interior of the atmospheric column (ATM) at an observation site in
Beijing can be calculated based on this algorithm. More detailed descriptions are
provided in our previous work (Xin et al., 2016).
Air temperature and relative and absolute humidity profiles were retrieved with a
microwave radiometer (hereinafter referred to as MWR) (RPG-HATPRO-G5 0030109,
Germany). The MWR produces profiles with a resolution ranging from 10-30 m up to
0.5 km, profiles with a resolution ranging from 40-70 m between 0.5 and 2.5 km and
profiles with a resolution ranging from 100-200 m from 2 to 10 km at a temporal
resolution of 1 s. More detailed information of the RPG-HATPRO-type instrument can
be found at http://www.radiometer-physics.de (last access: 4 June 2020). Vertical wind
speed and horizontal wind vector profiles were obtained by a 3D Doppler wind lidar
(Windcube 100s, Leosphere, France). The wind measurement results have a spatial
resolution ranging from 1-20 m up to 0.3 km and a spatial resolution of 25 m from 0.3
to 3 km, at a temporal resolution of 1 s. More instrument details can be found at
www.leosphere.com (last access: 4 June 2020). A ceilometer (CL51, Vaisala, Finland)
was adopted to acquire atmospheric backscattering coefficient (BSC) profiles. The
CL51 ceilometer digitally receives the return backscattering signal from 0 to 100 μs
and provides BSC profiles with a spatial resolution of 10 m from the ground to a height
of 15 km. The ABLH was further identified by the sharp change in the negative gradient
of the BSC profile (Münkel et al., 2007) and a detailed information is reported in
previous studies (Tang et al., 2015, 2016; Zhu et al., 2018). A CIMEL sun-photometer
(CE318, France), a multichannel, automatic sun-and-sky scanning radiometer (Gregory
2011), was used to observe the AOD, and the AOD at 500 nm is adopted in this paper.
The real-time hourly mean ground levels of $PM_{2.5}$ (particulate matter with aerodynamic
diameter less than or equal to 2.5 μm) and $PM_{10}$ (particulate matter with aerodynamic
diameter less than or equal to 10 μm) were downloaded from the China National
Environmental        Monitoring        Center        (CNEMC)        (available        at
http://106.37.208.233:20035/, last access: 4 June 2020).



The virtual potential temperature ($\theta_v$) and pseudoequivalent potential temperature
($\theta_{se}$) are calculated with Eqs. (1) and (2), respectively:
$\theta_v = T(1 + 0.608q)(\frac{1000}{P})^{0.286}$ (1)
$\theta_{se} = T(\frac{1000}{P})^{0.286} exp (\frac{r_s L_v}{C_{pd}T})$ (2)
where $T$ is the air temperature, $q$ is the specific humidity, $p$ is the air pressure, $r_s$ is the
saturation mixing ratio, $Lv$ is the latent heat of vaporization at $2.5 \times 10^6$ J kg$^{-1}$ and $C_{pd}$ is
the specific heat of air at 1005 J kg$^{-1}$ K$^{-1}$. All the relevant parameters can be calculated
from the temperature and humidity profile data obtained with the MWR, and the values
of $\theta_v$ and $\theta_{se}$ at different altitudes can be then further obtained. The hourly TKE is
calculated as:
TKE $= 0.5 \times (\delta_u^2 + \delta_v^2 + \delta_w^2).$ (3)
The one-hour vertical velocity standard deviation ($\delta_w^2$) and one-hour horizontal wind
standard deviation ($\delta_u^2$; $\delta_v^2$) are calculated with Eqs. (4), (5) and (6), respectively:
$\delta_w^2 = \frac{1}{N-1}\sum_{i=1}^{N}(w_i - \overline{w})^2$ (4)
$\delta_u^2 = \frac{1}{N-1}\sum_{i=1}^{N}(u_i - \overline{u})^2$ (5)
$\delta_v^2 = \frac{1}{N-1}\sum_{i=1}^{N}(v_i - \overline{v})^2$ (6)
where N is the number of records per hour, $w_i$ is the $i_{th}$ vertical wind velocity (m s$^{-1}$),
$u_i(v_i)$ is the $i_{th}$ horizontal wind speed (m s$^{-1}$), $\overline{w}$ is the mean vertical wind speed (m
s$^{-1}$), and $\overline{u}$ ($\overline{v}$) is the mean horizontal wind speed (m s$^{-1}$) (Banta et al., 2006; Wang et
al., 2019).
**3 Results and discussion**
**3.1 General haze episodes over Beijing in winter**
It is well known that severe air pollution episodes frequently occur in Beijing during
autumn and winter (Jin-Xiang, 2007; Zhang et al., 2017). Two-month PM concentration
data from Beijing in the winter of 2018 were collected. As expected, during this time,
Beijing experienced severe and frequent haze pollution episodes with two heavy
episodes in which the maximum hourly PM$_{2.5}$ concentration reached ~200 µg m$^{-3}$ and six
general episodes in which the PM$_{2.5}$ mass concentration ranged from ~100-150 µg m$^{-3}$



(Fig. S2(a)). Although the air pollution process is variable and complicated, it is worth
stating that the haze pollution in Beijing in winter can be generally classified as two
kinds of patterns, as shown in Fig. S2(b). For all haze episodes ①-⑦, the $PM_{2.5}$ mass
concentration slowly increased in the afternoon of the first day, followed by a secondary
maximum in the early morning and a maximum at midnight of the second day. In
comparison to the processes of ④-⑦, where the $PM_{2.5}$ mass concentration sharply
decreased to <25 μg m$^{-3}$ in the early morning of the third day, during periods ①-③,
however, the highest $PM_{2.5}$ mass concentration (~100-200 μg m$^{-3}$) was observed on the
third day, which disappeared on the fourth day. As previously reported, transport,
physical and chemical transformation and boundary layer structure (local
meteorological conditions) are central to the determination of the amount and type of
pollutant loading. The suspended particles in ④-⑦ were subjected to dispersal,
controlled by the atmospheric motion (wind and turbulence) on the third day. The
particles during periods ①-③ continued to accumulate and were therefore highly
related to the specific ABL status. To investigate the possible reasons for the different
variation trends of haze episodes ①-③ and ④-⑦, in the next section, we will mainly
focus on the ABL structure (local meteorological conditions) considering transport and
physical and chemical transformation.
**3.2 Qualitative analysis of the interaction between particulate matter and**
**boundary layer structure**

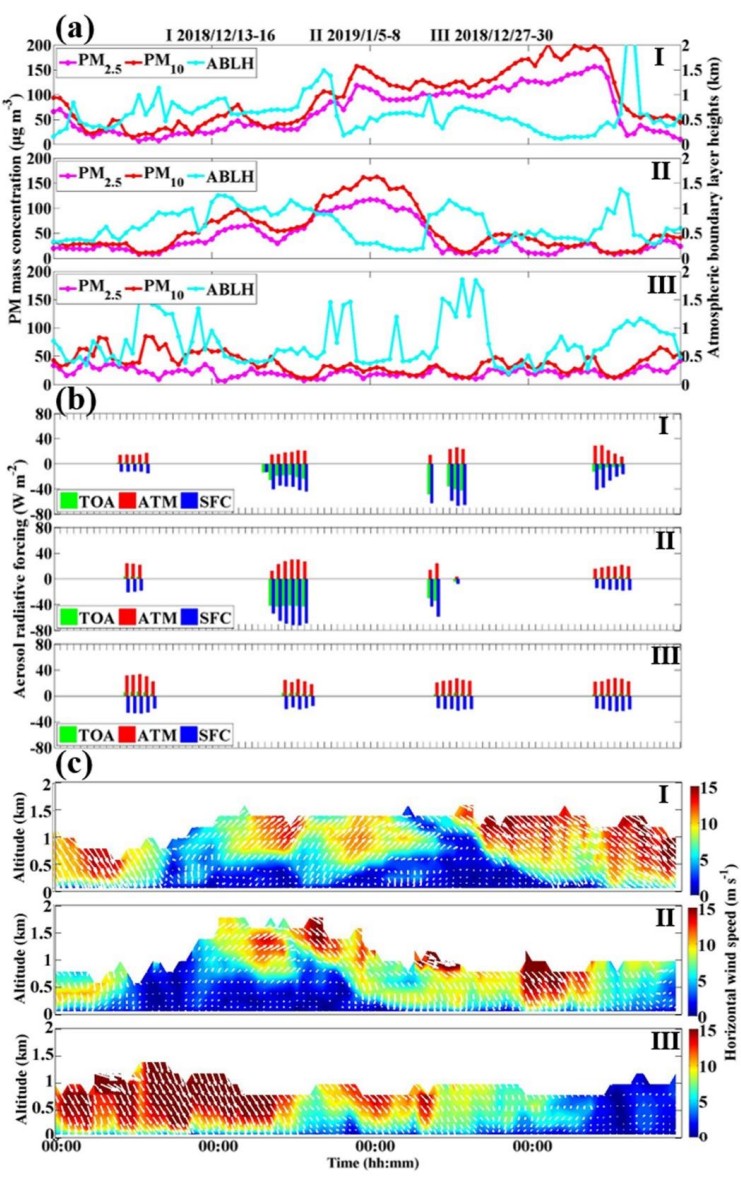


Figure 1. Temporal evolution of (a) the PM mass concentration and atmospheric

boundary layer height ($PM_{2.5}$: solid pink lines; $PM_{10}$: solid red lines; ABLH: solid blue

lines), (b) aerosol radiative forcing at the top (TOA; green bars), surface (SFC; blue

bars) and interior of the atmospheric column (ATM; red bars), and (c) horizontal wind

vector profiles (shaded colors: wind speeds; white arrows: wind vectors) during the

typical haze pollution episodes of I (2018/12/13-16) and II (2019/1/5-8) as well as the
typical clean period of III (2018/12/27-30).

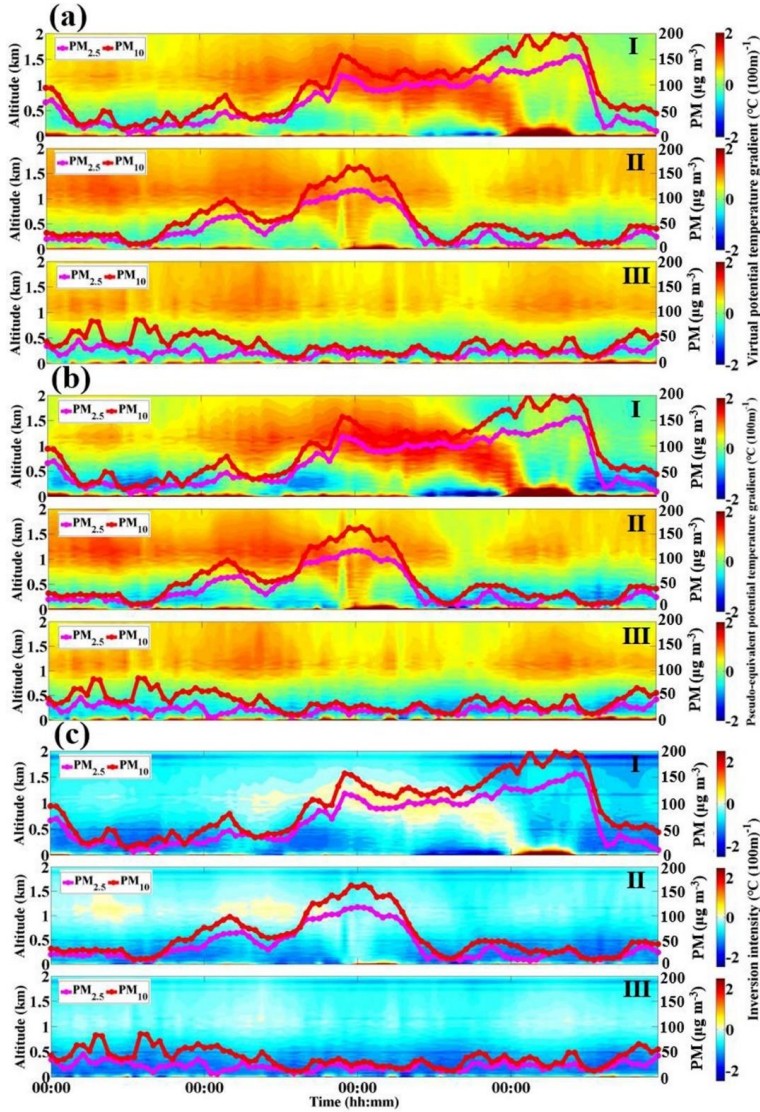


Figure 2. Temporal variation in the vertical profiles of (a) the virtual potential
temperature gradient ($\partial\theta_v/\partial z$), (b) pseudoequivalent potential temperature gradient
($\partial\theta_{se}/\partial z$) and (c) temperature inversion phenomenon (shaded colors: inversion intensity)
during the typical haze pollution episodes of I (2018/12/13-16) and II (2019/1/5-8) as
well as the typical clean period of III (2018/12/27-30).

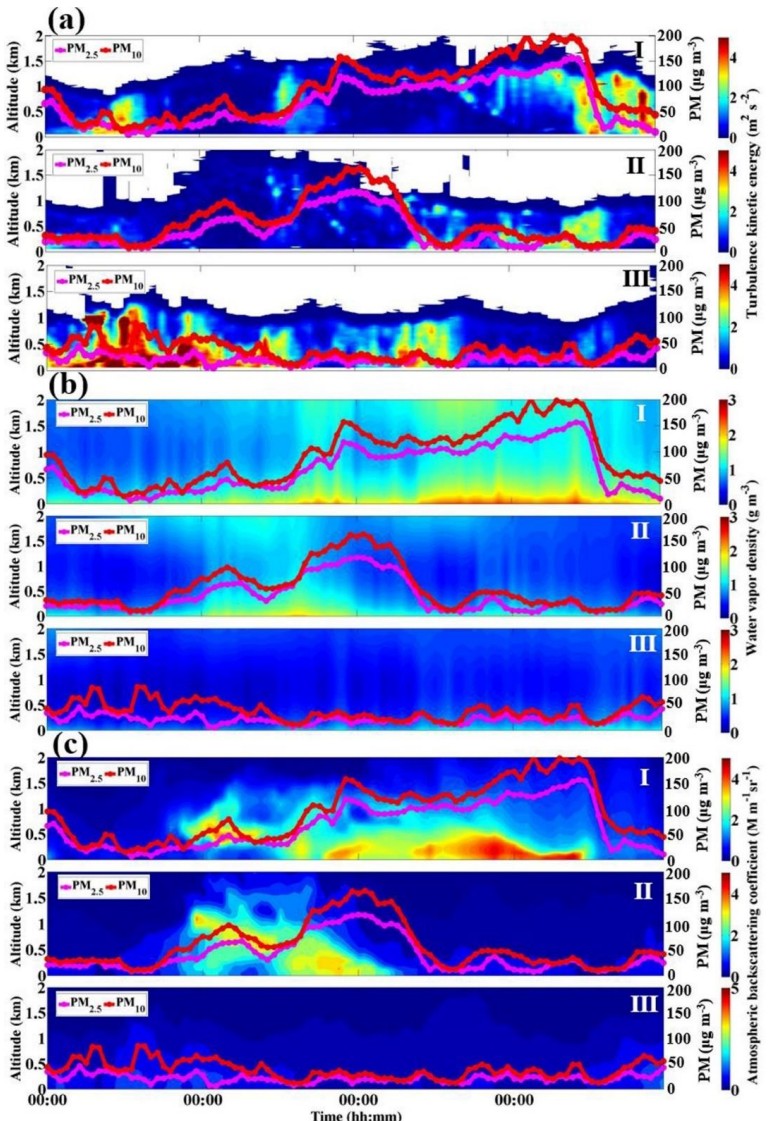


Figure 3. Temporal variation in the vertical profiles of (a) the turbulent activity (shaded
colors: TKE), (b) atmospheric humidity (shaded colors: vapor density) and (c) vertical
distribution of suspended particles (shaded colors: BSC) during the typical haze
pollution episodes of I (2018/12/13-16) and II (2019/1/5-8) as well as the typical clean
period of III (2018/12/27-30).
As described in the previous section, although not exactly the same, the haze episodes
followed two different kinds of variation trends. The specific reason for this finding





will be systematically analyzed in this section. To better illustrate the two different haze
pollution patterns, a typical clean period will be considered as a control. The typical air
pollution episodes of I (2018/12/13-16) and II (2019/1/5-8) as well as the typical clean
period of III (2018/12/27-30) are chosen as examples for analysis. Numerous studies
have reported that the original explosive growth of PM is caused by pollution transport
under southerly winds (Ma et al., 2017; Zhao et al., 2019; Zhong et al., 2018). In this
study, the action of southerly winds on the air pollution in Beijing was presented more
clearly as the distribution of the horizontal wind vectors extending to heights of 1-1.5
km (equivalent to the entire ABL) was obtained by the Windcube 100s lidar (Fig. 1(c)).
On the 1st day of episodes I and II, the atmosphere layer up to ~1 km in height was
controlled by strong and clean north winds, exactly like clean period III. Clearly, no
pollution transport occurred, and the PM and ARF levels were equivalent to those on a
clean day (Figs. 1(a)-(b)). The atmospheric backscattering coefficients throughout the
ABL during the three episodes only ranged from ~0-1.5 M m$^{-1}$sr$^{-1}$ (Fig. 3(c)). From the
evening of the 1st day to the forenoon of the 2nd day, strong southerly winds blew across
Beijing during both episodes I and II, with the wind speed increasing with the height,
reaching ~5-15 m s$^{-1}$ at an atmosphere of about 0.5-1.5 km. The ABL during clean
episode III was still dominated by north winds. Sensitive to the change in wind direction
from north to south, the PM mass concentration progressively increased from a fairly
low level to ~50 μg m$^{-3}$. Moreover, the BSCs sharply increased to ~3 Mm$^{-1}$ rd$^{-1}$ and was
concentrated at altitudes from ~0.5-1 km, which further stressed the effects of southerly
transport on the original growth of the PM mass concentration over Beijing. With winds
originating from the wetter south, compared to the low humidity during clean episode
III, the air humidity in Beijing during this time notably increased with the vapor density
ranging from ~1.5-2 g m$^{-3}$ during both episodes I and II (Fig. 3(b)). During the
remainder of the 2nd day, the PM mass concentration continued to increase with south
winds blowing and reached its highest level at midnight with a PM$_{2.5}$/PM$_{10}$ mass
concentration of ~110/150 μg m$^{-3}$ during both episodes I and II. The highest BSC values
mainly occurred from the ground to a height of 1 km at this time, implying that a portion





of the suspended particles was pushed down to the near-surface. Noteworthily,
regardless of the wind field, the atmospheric stratification states during this rising phase
changed more notably. Before southerly wind transport occurred, the evolution of the
stability indicator ($\partial\theta_v/\partial z$; $\partial\theta_{se}/\partial z$) profiles during episodes I and II was analogous to that
during episode III (Figs. 2(a)-(b)). The stratification states at the different heights (0-1
km) were either unstable or neutral, with negative or zero $\partial\theta_v/\partial z$ values, respectively,
whereby no clear nor strong temperature inversion phenomenon occurred in the lower
atmosphere layer (Fig. 2(c)). The corresponding ABLHs were the same (Fig. 1(a)).
However, the atmospheric stratification from ~0.5-1 km during episode I and from 0-1
km during episode II became quite stable during the PM increase period, with positive
values of $\partial\theta_{se}/\partial z$ and almost no turbulent activity (TKE: ~0 $m^2\ s^{-2}$) (Fig. 3(a)). In contrast
to an increased ABLH during clean period III, the ABLHs during episodes I-II sharply
decreased. Considering that aerosol scattering and absorbing radiation could modify the
temperature stratification (Li et al., 2010; Zhong et al., 2018), the aerosol radiation
effect is too weak at a low PM level to change the latter, which defines the atmospheric
stability. With the elevated PM level due to southerly transport, ARF also increased,
with SFC (ATM) reaching ~-40 (~20) W $m^{-2}$ and ~-75 (~30) W $m^{-2}$ during episodes I
and II, respectively. Less radiation reaching the ground and more heating the
atmosphere above the ground, and in comparison to clean episode III, the atmospheric
stratification during episodes I and II was altered. Besides, TOA has an analogous
variation trend with SFC, increasing from quite low values to ~-20 W $m^{-2}$ and ~-45 W
$m^{-2}$ during episodes I and II, respectively. It further clarified the high scattering effect
of aerosols with the elevated PM level. The suspended particles carried by southerly
transport originally occurring at high altitudes were restrained from vertically spreading
and gradually sank due to gravity and accumulated near the surface. This stable
stratification has a certain impact on aggravating haze pollution.

271        It is salient to note that the haze evolution trends during episodes I and II were

basically consistent so far, corresponding to a similar ABL structure. Nevertheless, the
north winds (~10-15 m $s^{-1}$) during episode II, which only blew above the ABL (>1 km)



at midnight of the 2$^{nd}$ day, gradually spread downward and controlled the whole
boundary layer on the 3$^{rd}$ day. Moreover, the south wind, which once was strong and
filled the boundary layer on the 2$^{nd}$ day during episode I, gradually decelerated over
time from the ground to high altitudes on the 3$^{rd}$ day. The wind field is critical with
respect to horizontal dispersion in the boundary layer; thus, the strong, clean and dry
north winds during episode II greatly diffused the already accumulated particles first,
where the $PM_{2.5}$ mass concentration decreased from ~100 to ~50 μg m$^{-3}$. The ARF
decreased to the same level as that during clean period III, and with solar radiation
heating the ground at noon on the 3$^{rd}$ day, the positive sensible heat flux (upward heat
transfer) eliminated the previous night's temperature structure. The temperature
stratification became similar to that on clean day III with a similar increase in ABLH.
Thus, an unstable/neutral atmospheric state with a TKE of ~2 m$^2$ s$^{-2}$ was also conducive
to the vertical spread of materials, which were replaced with cleaner air from above. In
response, the PM mass concentration (BSC) and air humidity during episode II
gradually decreased and reached the same level as those during episode III. Conversely,
the whole ABL (0-1 km) was controlled by calm/light winds during episode I on the 3$^{rd}$
day. On account of the calm/light winds, the horizontal wind shear sharply decreased,
resulting in a decline in the intensity of mechanical turbulence. In the absence of an
existing high PM mass concentration, strong ARF would continue to notably cool the
ground and heat the aerosol layer, keeping the atmospheric stratification stable and thus
decreasing the intensity of thermal turbulence. As can be seen in Fig. 1(b), SFC and
TOA further increased up to ~-40 W m$^{-2}$ and ~-75 W m$^{-2}$, respectively, with ATM
remaining high (~25 W m$^{-2}$). The ABLH barely changed on the 3$^{rd}$ day and maintained
a lower altitude in the afternoon of the 4$^{th}$ day. Therefore, a rather stable atmosphere
extended from ~0.3-0.5 km to ~1.5 km on the 3$^{rd}$ day and from the ground to heights of
~0.3 km in the afternoon of the 4$^{th}$ day (Figs. 2(a)-(c)). The quite low TKE was highly
consistent with the atmospheric stability stratification. Since the stable stratification
acted as a lid at altitudes from 0.5-1.5 km, downward momentum transport would be
blocked, further explaining the calm/light winds in the lower atmosphere layer. In the





afternoon of the 4th day, it is worth noting that above the stable atmospheric stratification
(0-0.3 km altitude), a relatively strong horizontal wind shear occurred corresponding to
a TKE of ~1-2 $m^2$ $s^{-2}$. The accumulated particles near the surface were further inhibited
right below the stable atmosphere layer, as reflected by the BSC distribution. This
highlights the fact that a stable atmosphere with a weak turbulent activity was central
to pushing down the pollutant layer. The same work was exerted on the water vapor as
the air humidity at this time reached ~3 g $m^{-3}$ below an altitude of ~0.3 km,
accompanied by intense heterogeneous hydrolysis reactions at the moist particle surface
(Zhang et al., 2008), which further increased the PM mass concentration. In the
afternoon of the 4th day, north winds spread down to the whole ABL, which promoted
the horizontal and convective dispersion of pollutants and water vapor, and the PM
mass concentration therefore dropped to the same level as that on clean day III. With
quite low aerosol loading, the aerosol radiative effect was also quite weak and the ARF
dropped to dropped to the level of that on clean day III. In this section, through a
detailed contrastive analysis, we examined the potential reasons for the occurrence of
the two different patterns of haze pollution and found that the crucial point in
determining whether the PM mass concentration remained high or sharply decreased
was related to whether the boundary layer remained stable. The boundary layer stability
was in turn notably linked to the PM mass concentration and aerosol radiative effect.
**3.3 Quantitative analysis of the effect of particulate matter on the boundary layer**
**structure**



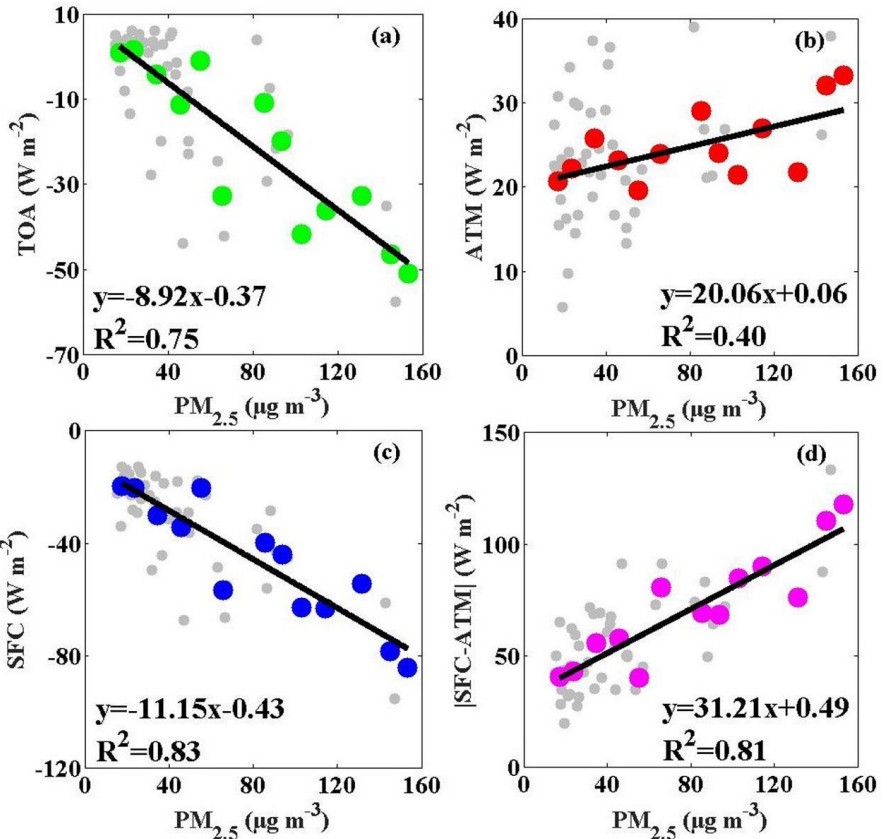

Figure 4. Scatter plots of the PM$_{2.5}$ mass concentration (x) versus aerosol radiative forcing at the surface (SFC; y; a), interior of the atmospheric column (ATM; y; b) and top of the atmospheric column (TOA; y; c) as well as the absolute difference of SFC and ATM (|SFC-ATM|; y; d), respectively (gray dots: daily data; other dots: mean data). The calculated daily data were collected over a two-month period in Beijing from 27 November 2018 to 25 January 2019. (The daily data means daily mean values of TOA, ATM, SFC and corresponding daily averaged PM$_{2.5}$ mass concentration. The mean PM$_{2.5}$ concentration were calculated at intervals of 10 μg m$^{-3}$ daily PM$_{2.5}$ concentration, then the mean TOA, ATM and SFC were obtained after the average of the corresponding daily TOA, ATM and SFC, respectively.).



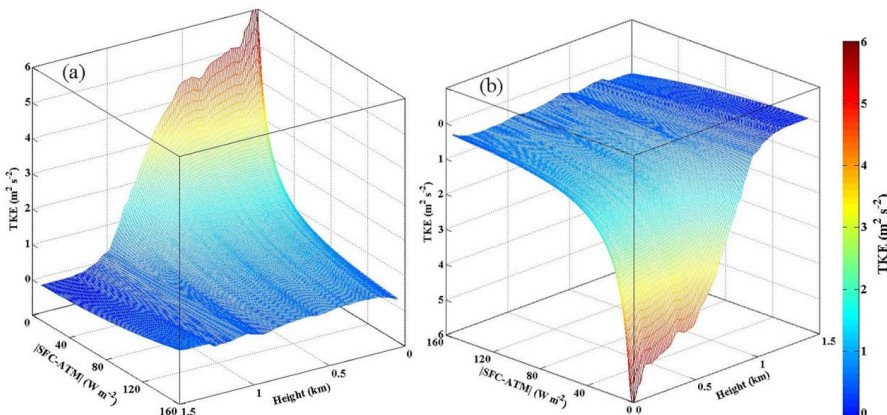

Figure 5. 3-D plot of the fitting relationship of the absolute difference in aerosol radiative forcing between the surface and interior of the atmospheric column (|SFC-ATM|; x) and turbulence kinetic energy (TKE; z) at the different altitudes (y) ((a) and (b) present different perspectives).

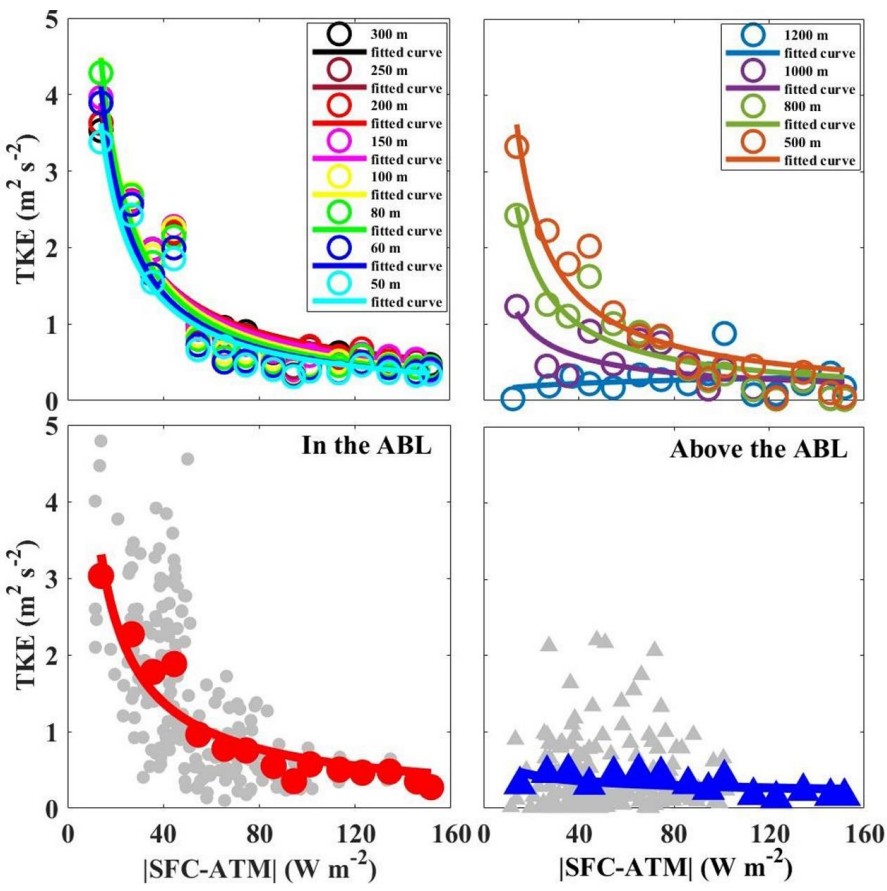

Figure 6. Scatter plots of the mean absolute difference of the aerosol radiative forcing at the surface and interior of the atmospheric column (|SFC-ATM|; x) versus the mean turbulence kinetic energy (TKE; y) at the different altitudes (the top row). Scatter plots of |SFC-ATM| (x) versus TKE (y) in the ABL and above the ABL (the bottom row; gray dots: hourly data; other dots: mean data). The hourly data were collected over a two-month period in Beijing from 27 November 2018 to 25 January 2019. (The hourly data means hourly mean values of |SFC-ATM| and corresponding hourly TKE. The mean |SFC-ATM| was averaged at intervals of 10 W m$^{-2}$ hourly |SFC-ATM|, then the mean TKE was obtained after the average of the corresponding hourly TKE.).

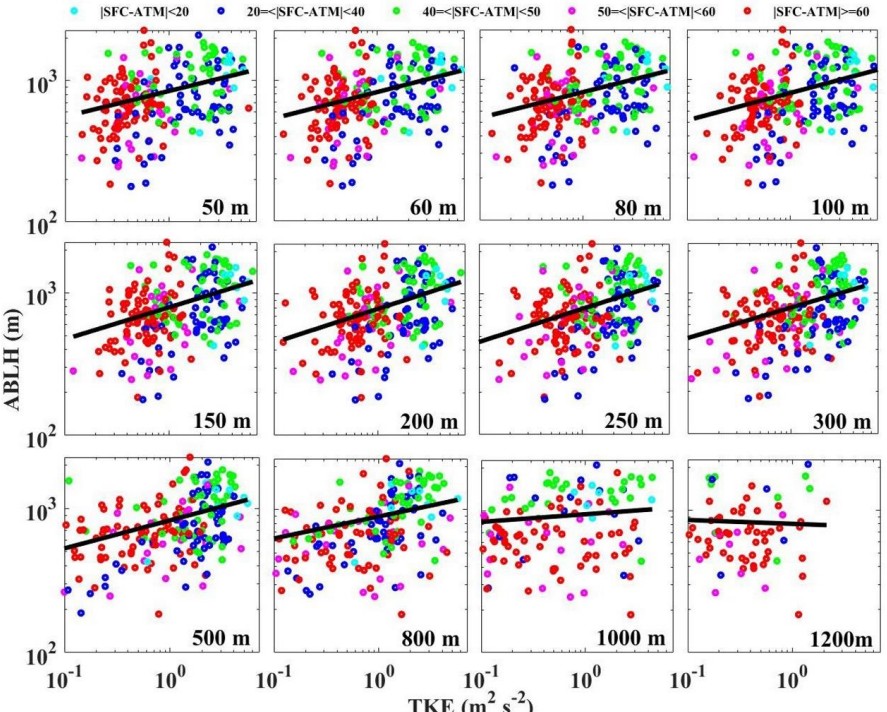

Figure 7. The atmospheric boundary layer height (ABLH; y) as a function of the turbulence kinetic energy (TKE; x) at the different altitudes and the aerosol radiative effect defined as |SFC-ATM| (color code). The calculated hourly data used above are collected over a two-month period in Beijing from 27 November 2018 to 25 January 2019.

Based on the contrastive analysis in the previous section, it was clear that the stable ABL structure played a critical role in the outbreak and maintenance of air pollution. It appeared that the increase in atmospheric stability suppressed pollution diffusion under a weak turbulence activity and low ABLH. Water vapor also greatly accumulated to a quite high level near the surface, further facilitating the formation of secondary aerosols. The evolution of ABL stability essentially occurred in response to the atmospheric temperature structure, as analyzed above, which was influenced by the strong aerosol radiation effect (Li et al., 2010; Andrews, 2000). The Archimedes buoyancy generated by the pulsating temperature field in the gravity field exerted negative work on the turbulent pulsating field with a stable ABL occurring, and the turbulence served as a





carrier for substance transport in the boundary layer, such as water vapor, heat and PM.
(Garratt et al., 1992). Generally, the ABL structure controlling pollutant dissipation
therefore greatly relies on the turbulent activity. Thus, in the following section, the ARF
and TKE were chosen as the key parameters to examine how PM affects and modifies
the boundary layer structure.

371        Figure 4 shows the relationship between the PM concentration and ARF. The

aerosol scattering effect results in less radiation reaching the ground and the top of the
atmospheric column, so the solar radiation levels reaching the ground and at the top of
the atmospheric column differ with or without ambient aerosols, thus making SFC and
TOA forcing. As shown in Figs. 4(a) and (c), SFC and TOA, respectively, were basically
proportional to the $PM_{2.5}$ concentration. With the increase in $PM_{2.5}$ concentration, the
solar radiation reaching the ground and at the top of the atmospheric column decreased,
corresponding to a cooling of the ground and top of the atmospheric column. ATM,
driven by aerosol absorption and representing a warming effect of aerosols on the
atmosphere layer, exhibited a positive correlation with the $PM_{2.5}$ concentration (see Fig.
4(b)). These results demonstrated that a higher $PM_{2.5}$ concentration would arouse a
stronger ARF, further inhibiting solar radiation from reaching the ground, thus more
notably heating the atmosphere layer. |SFC-ATM|, defined as the absolute value of the
difference between SFC and ATM, represents the combined action of aerosols on the
solar radiation reaching the aerosol layer and the ground. Larger values of |SFC-ATM|
indicate stronger aerosol scattering and/or absorption effects, further implying a more
significant temperature difference between the ground and the above atmosphere layer.
As expected, a positive linear correlation between |SFC-ATM| and $PM_{2.5}$ concentration
was found, as shown in Fig. 4(d).

390        As described in the above paragraph, there was a strong ARF under a high PM

loading, which markedly altered the atmospheric temperature structure, further
changing the ABL structure. It is necessary to determine the effect degree of ARF on
the boundary layer structure. Figure 5 shows the 3-D plots of the fitting relationship
between the hourly values of |SFC-ATM| and TKE at the different altitudes from





different perspectives. What stood out in Fig. 5(a) was the general decline in TKE with
respect to the growth of |SFC-ATM|. With increasing |SFC-ATM| value, the TKE value
at the different altitudes always decreased exponentially and approached zero below
~0.8 km. The notable exponential function between TKE and |SFC-ATM| explained
that a strong ARF would drastically change the boundary layer into highly stable
conditions characterized by a rather low TKE. The results above highlight the
nonnegligible impact of the aerosol radiative effect on the boundary layer structure,
especially during the haze episode under a high aerosol loading with a strong ARF. It is
well known that a larger net negative/positive SFC/ATM means a cooler/warmer the
ground/atmosphere would be. An increase in |SFC-ATM| implies the gradual
intensification of the ground cooling and/or atmosphere heating processes. It therefore
changed the atmospheric stratification into a gradually enhanced stable state, which was
characterized by increasingly weaker turbulence activities. Additionally, as shown in
Fig. 5(b), from another perspective, we can clearly identify a critical point of the |SFC-
ATM| effects on TKE in the low layers. In particular, TKE decreased with increasing
|SFC-ATM| and hardly changed when |SFC-ATM| exceeded the critical point. To define
the critical point, we generated scatter plots of the average |SFC-ATM| and TKE at
several altitudes, as shown in Figs. 6(a)-(b). The |SFC-ATM| had mean values of 10-20,
20-30, …, and 150-160 W m$^{-2}$, and the corresponding mean TKE values were further
calculated. The scatter plots of the unaveraged hourly data are shown in Fig. S3, and
the fitting functions are listed in Table S1. Depending on the maximum curvature of the
exponential curve (Silvanus and Gardner, 1998), a critical point should exist. With the
mean TKE and |SFC-ATM| values on the exponential curve, we found that once the
aerosol radiative effect defined by |SFC-ATM| exceeded 50-60 W m$^{-2}$ (average of ~55
W m$^{-2}$), the TKE sharply decreased from ~2 m$^2$ s$^{-2}$ to lower than 1 m$^2$ s$^{-2}$. This means
that a high aerosol loading with a |SFC-ATM| value higher than ~55 W m$^{-2}$ would
change the boundary layer from the unstable state to the extremely stable state in a short
time, and further increasing |SFC-ATM| would barely modify the ABL structure. This
result can provide useful information to explain why air pollution is sometimes



aggravated under a stable ABL and sometimes not. The average aerosol radiative
forcing (|SFC-ATM|) value of ~55 W m$^{-2}$ can be defined as the threshold of the ARF
effects on the ABL structure, which could provide useful information for relevant model
simulations, atmospheric environment improvement measures and relevant policies. In
addition, as shown in Figs. 5 and 6, the exponential relationship between TKE and
|SFC-ATM| was notable in the low layers and gradually deteriorated with increasing
altitude. On average, the exponential relationship was notable in the ABL and almost
disappeared above the ABL (Figs. 6(c) and (d)). Considering that aerosols are mainly
concentrated below the lower atmosphere, contributing the most to the SFC and ATM
forcing, which further confirmed, the considerable change in atmospheric stratification
caused by aerosols indeed existed and mainly occurred in the lower layers.

435        From the previous discussion, it is clear that a strong aerosol radiative effect

markedly affected the turbulent activity and modified the boundary layer structure. As
many studies have reported, the ABLH is an important meteorological factor that
influences the vertical diffusion of atmospheric pollutants and water vapor (Stull, 1988;
Robert and Aron, 1983). The following is an examination of the relationship among the
turbulent activity, ARF and ABLH to illustrate the change in ABLH in response to ARF.
Figure 7 shows the ABLH as a function of the TKE and |SFC-ATM| at the different
altitudes. It was apparent from this figure that a positive correlation exists between TKE
and ABLH. As the turbulent activity became increasingly weaker, the corresponding
boundary layer height gradually decreased, which was in response to the gradual
increase in |SFC-ATM|. Similar to the relationship between the turbulent activity and
aerosol radiative effect, as shown in Fig. 6, the relationship among these aspects was
much stronger below 300 m and almost disappeared above 800 m. This further
addressed the fact that the change in boundary layer height was attributed to the
turbulence activity variation, which stemmed from the aerosol radiative effect.

450        Thus far, this section has demonstrated that the aerosol loading with aerosol

radiative effects impacted the turbulent activity, changed the boundary layer height and
thus modified the boundary layer structure. On the other hand, it is now necessary to



explain how the renewed boundary layer structure modifies the $PM_{2.5}$ concentration. As
shown in Figs. S4(a)-(b), the ABLH as an independent variable has an impact on the
ambient water vapor in the ABL at some degree. There was a steady increase in the
ambient humidity with decreasing ABLH, where absolute humidity (AH) and relative
humidity (RH) were projected to decrease to ~3 g m$^{-3}$ and ~60%, respectively, with the
ABLH decreasing below ~500 m. With the increase in ambient humidity, a marked rise
in $PM_{2.5}$ concentration occurred, as shown in Figs. S4(c)-(d). Once AH and RH
exceeded ~3 g m$^{-3}$ and ~60%, respectively, the $PM_{2.5}$ concentration reached ~100 μg m$^{-}$
$^{3}$. The results above indicate that with a fairly low boundary layer height, water vapor
accumulated near the surface, and particles tended to hygroscopic grow, resulting in
secondary aerosol formation in a high-humidity environment, further increasing the
$PM_{2.5}$ concentration. As shown in Fig. S4(e), with the level off of the ABLH, the $PM_{2.5}$
mass concentration increased exponentially and reached a high value. The exponential
relationship was similar to that between the ambient humidity and ABLH, which
revealed that the explosive growth of the $PM_{2.5}$ concentration under a low ABLH was
largely driven by intense secondary aerosol formation and hygroscopic growth at a high
ambient humidity.

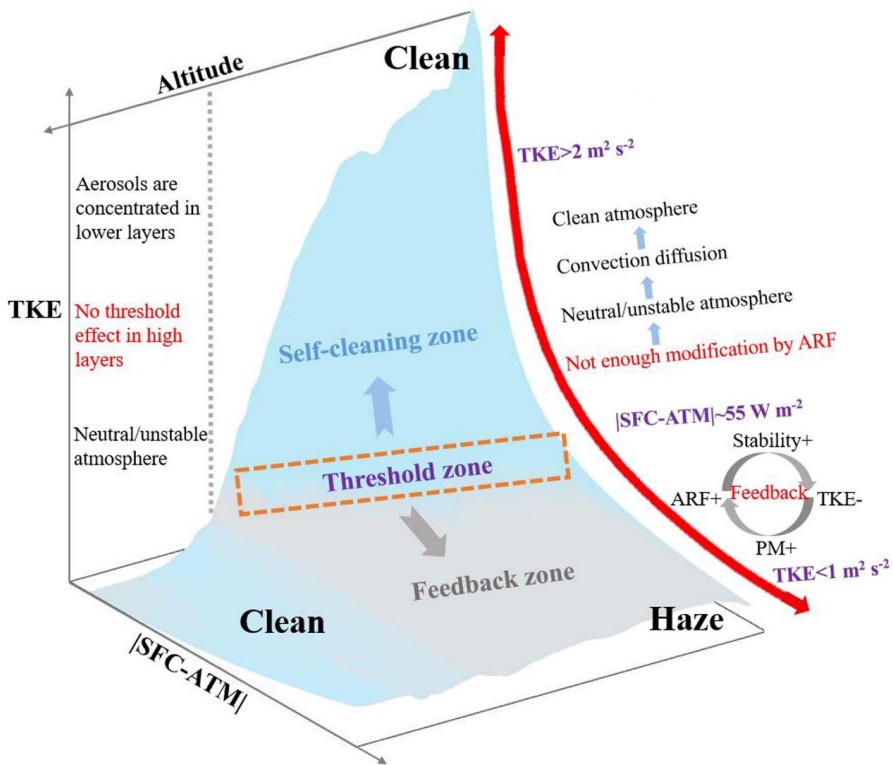


Figure 8. Schematic diagram of the interaction between the aerosol radiation forcing (ARF) and
boundary layer structure (|SFC-ATM|: the mean absolute difference of the aerosol radiative
forcing at the surface and interior of the atmospheric column; TKE: the mean turbulence kinetic
energy).
**4 Conclusion**
By analyzing the two-month haze conditions in Beijing in winter, we found that haze
pollution underwent two different variation patterns, namely, the same trends on the
first two days, and on the next days, one haze pattern went through a continuing
outbreak, while the other haze pattern exhibited notable diffusion. Considering
equivalent emissions, this has raised important questions about whether and how local
meteorological conditions, as well as the boundary layer structure, impacted/caused this
difference. The results of a contrastive analysis qualitatively showed that the crucial
point in determining whether the PM concentration remained very high or sharply
decreased was related to whether the boundary layer structure (i.e., stability and TKE)





satisfied relevant conditions. As previous studies reported (Huang et al., 2018; Liu et
al., 2018; Zhong et al., 2018) and was confirmed in this paper, the extremely stable
stratification with positive $\partial\theta_{se}/\partial z$ values and a low TKE was the premise of the outbreak
of haze pollution. However, it appeared that the change/state of the boundary layer
structure was in turn strongly linked to the PM mass concentration and ARF, and we
further quantitatively evaluated the effect of ARF on the boundary layer structure. The
Fig. 8 emerging from the foregoing observation analysis is one where ARF modifies
the boundary layer structure and aggravates haze pollution. The aerosol effects on the
atmospheric stratification depend on the reduced radiation reaching the ground due to
aerosol scattering and absorbing radiation in the atmosphere (Dickerson et al., 1997;
Stone et al., 2008; Wilcox et al., 2016). First, we found that there existed a positive
linear relationship between |SFC-ATM| and $PM_{2.5}$ concentration, which means that
strong aerosol scattering and/or absorption effect occurred during the heavy haze
episodes could arouse significant temperature differences between the ground and the
above atmosphere layer. Previous studies revealed that black carbon solar absorption
suppresses turbulence in the ABL (Wilcox et al., 2016), however, we found that the
TKE value at the different altitudes always decreased exponentially and approached
zero with increasing |SFC-ATM|, which was significant in the lower atmosphere layer
and became gradually worse with increasing altitude. Then, we confirmed that the
decrease in boundary layer height was attributed to the reduction in turbulence activity,
stemming from the intensification of the aerosol radiative effect. Thus, with the increase
in PM mass concentration, the temperature stratification at the different heights in the
boundary layer gradually shifted from the normal state in which the turbulence activity
is strong and the boundary layer height is quite high to the abnormal state that
suppresses turbulence development and decreases the boundary layer height. The ARF
effects on atmospheric stratification were more significant in the lower layer and
disappeared above the boundary layer, which also confirmed that the stronger ARF
from the aerosol layer would indeed change the boundary layer into the considerably
stable state characterized by a rather low TKE. The change in ARF is linear due to the





PM concentration; however, the influence of ARF on the boundary layer structure is
nonlinear. Based on the exponential relationship, the threshold of the ARF effects on
the boundary layer structure has been determined for the first time in this paper, which
highlighted that once the ARF exceeded a certain value, the boundary layer structure
would quickly stabilize and thereafter changed little with increasing ARF. The
discovery of this threshold further quantifies the feedback mechanism of the ARF on
the boundary layer stability, and the occurrence of this feedback mechanism is directly
related to the degree of pollution. The threshold of the ARF effects on the boundary
layer stability can provide useful information for relevant atmospheric environment
improvement measures and policies, such as formulating the objectives for phased air
pollution control. When the $PM_{2.5}$ concentration is controlled with the ARF below the
threshold, the self-purification capacity of the atmosphere can effectively dilute and
diffuse pollutants. The pollution concentration decreases rapidly, the weak ARF and
free convection of the atmosphere produce a virtuous cycle, and the atmosphere
maintains a high efficiency of self-purification. In contrast, when the $PM_{2.5}$
concentration increases with an ARF exceeding the threshold value, this further
stabilizes the boundary layer, and the atmospheric environmental capacity rapidly
decreases, especially near the stratum. The $PM_{2.5}$ concentration further increases,
aggravating haze pollution. In the process of air pollution control, there is a nonlinear
relationship between the $PM_{2.5}$ concentration in the atmosphere and the emission
control amount of the source, which is also the most difficult stage for the control of
polluted areas. On all accounts, this study provides the first comprehensive assessment
of the interaction between PM and boundary layer structure through qualitative and
quantitative analysis. The estimation of the ARF effects on the boundary layer structure
can also be adopted as a reference in model studies.
**Data availability**
The surface $PM_{2.5}$ & $PM_{10}$ and other trace gases observation data used in this study can
be accessed from http://106.37.208.233:20035/ (last access: 4 June 2020). Other
datasets can be accessed upon request to the corresponding author.



**Author contribution**

ZD performed the research and wrote the paper. XJ provided writing guidance, revised and polished the paper. GC performed the SBDART model. QJ and WY GC contributed to discussions of results. TG and MY designed the experiments and DL, WX, LG and MY carried them out. All the authors have made substantial contributions to the work reported in the manuscript.

**Competing interests**

The authors declare that they have no conflict of interest.

**Acknowledgments**

This study was supported by the Ministry of Science and Technology of China (grant number 2016YFC0202001), the CAS Strategic Priority Research Program (XDA23020301) and the National Natural Science Foundation of China (grant number 41375036). The authors are thankful for the data support from the National Earth System Science Data Sharing Infrastructure, National Science and Technology Infrastructure of China (available at http://www.geodata.cn, last access: 4 June 2020).

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
