# Peer review of "The impact threshold of the aerosol radiative forcing on the boundary layer structure in the pollution region"

_Atmospheric Chemistry and Physics, 2020_

## Author Comment (AC1) · 26 Aug 2020

The corrected Abstract is shown as below: Abstract: Recently, there has been increasing interest in the relation between particulate matter (PM) pollution and atmospheric boundary layer (ABL) structure. This study aimed to qualitatively assess the interaction between PM and ABL structure in essence, and to further quantitatively estimate the effects of aerosol radiative forcing (ARF) on the ABL structure. Multi-episode contrastive analysis stated the key to determining whether haze outbreak or dissipation was the ABL structure (i.e., stability and turbulence kinetic energy (TKE)) satisfied relevant conditions. However, it seemed that the ABL structure change was in turn highly

related to the PM level and ARF. |SFC-ATM| (SFC and ATM is respectively the ARF at the surface and interior of the atmospheric column) is the absolute difference between ground and atmosphere layer ARFs, and the change in |SFC-ATM| is linearly related to the PM mass concentration. However, the influence of ARF on the boundary layer structure is nonlinear. With increasing |SFC-ATM|, the TKE level exponentially decreased, which was notable in the lower layers/ABL but disappeared above the ABL. Moreover, the threshold of the ARF effects on the ABL structure was determined for the first time, namely, once |SFC-ATM| exceeded $\sim 55$ W m-2, the ABL structure would quickly stabilize and would thereafter change little with increasing ARF. The threshold of the ARF effects on the boundary layer structure could provide useful information for relevant atmospheric environment improvement measures and policies, such as formulating the objectives of phased air pollution control.

---

## Author Comment (AC2) · 26 Aug 2020

Line 15 in the Abstract "However, this has yet to be fully understood because most studies have been superficial" is inappropriate which was deleted and explained in our previous minor revision. However we forgot to upload the corrected Abstract, thus, we make a comment here to state it.

---

## Referee Comment (RC1) · Anonymous Referee #1 · 21 Oct 2020

The interaction between aerosols and the boundary layer is a hot topic in the study of the formation mechanism of air pollution in polluted areas. The aim of this paper is to evaluate the fundamental interaction between PM and ABL structure and to further quantitatively estimate the effect of aerosol radiative forcing (ARF) on ABL structure. The paper addressed relevant scientific questions and presented novel concepts, ideas and tools. The scientific methods and assumptions were almost valid and clearly outlined,  so that substantial conclusions were reached. The description of experiments and calculations were almost complete and precise to allow their reproduction by fellow scientists except some points. I think the manuscript could be considered to be accepted after major revision.

**Major comments:**

1. Seven cases spanning a period of 2 months were selected in the paper to discuss the threshold value of the effect of aerosol radiative forcing on the boundary layer structure of the contaminated areas. Are these cases representative, and would the thresholds change in other cases?

2. With only a finite number of points in Fig. 4, does the current fitting relationship pass the significance test?

3. In Fig. 1b, the results for aerosol radiative forcing have values only for individual moments of the day, and a detailed explanation of how they relate to hourly variations in atmospheric conditions and PM concentrations is needed.

4. What is the physical mechanism by which |SFC-ATM| affects the threshold of atmospheric stability?

5. When calculating TKE, why a one-hour wind standard deviation was chosen rather than a half-hour or two-hour standard deviation? In lines 141-152, the temporal and spatial scales of TKE need to be clarified.

6. Fig. 1 is of low quality and should be improved. In Fig. 1-(a)-III, why does the PM not increase with decreasing ABLH?

7. In Figure 1-(b) I and II, the TOA varied significantly. What is the reason?

8. there are very interesting results for PM and temperature in Figure 2. What are the diurnal characteristics of the potential temperature? Does potential temperature

affect the diurnal concentration of $PM_{2.5}$?

9.  In Figure 4, other dots represent mean data. How is it calculated?

10. The empirical relationships of TKE and |SFC-ATM| are very interesting in Figure 6 9 (left upper panel). It established the thermodynamic relationship between ARF and TKE by using the measured data. Why does the fitting relationship fit so well below 300 meters?

11. The ARF threshold is about 55 $Wm^{-2}$. What about the concentration of $PM_{2.5}$? Is it possible to derive a threshold concentration for $PM_{2.5}$ based on current observational relationships. the $PM_{2.5}$ threshold would be a very meaningful target for air pollution control.

12. The review of aerosol radiative forcing in the introduction needs to be strengthened.

13. Conclusion needs to be subdivided and further simplified.

14. In Figure 8, TKE >2 $m^2S^{-2}$, |SFC-ATM| ~55 W $m^{-2}$ . Are these thresholds generalizable?

**Minor comments:**

English writing should be polished. Some sentences was hard to read.

1.  e.g.   line 18-20 "Multi-episode contrastive analysis stated the key to determining whether haze outbreak or dissipation was the ABL structure (i.e., stability and turbulence kinetic energy (TKE)) satisfied relevant conditions."   Should be "Multi-period comparative analysis indicated that the key to determining whether the haze outbreak or dissipation occurs is whether the ABL structure (i.e., stability and turbulent kinetic energy (TKE)) satisfies the relevant conditions."

2.  Line 22-23. "SFC and ATM is respectively the ARF at the surface and interior of the atmospheric column" should be "SFC and ATM are the ARFs at the surface and interior of the atmospheric column, respectively."

3.  Line 37-38. (Li et al., 2020; Xu et al., 2019), should be cited at the end of this sentence.

4.  Line 316   two "dropped to "

---

## Referee Comment (RC2) · Anonymous Referee #2 · 14 Nov 2020

**Review comments on "The impact threshold of the aerosol radiation forcing on the boundary layer structure in the pollution region" by Zhao et al., 2020.**

1. The authors attempted to propose a parameter, $|SFC - ATM|$ for quantification of the impact of aerosol radiative forcing (ARF) on the atmospheric boundary layer (ABL) structure. Why did the author use the ARF of the interior of the atmosphere column ($ATM$) rather than the ARF in the ABL since most of aerosols or particulate matters are trapped in the atmospheric boundary layer?

2. Impact of ARF on reduction of surface-reaching short-wave radiation and heating/cooling of the atmosphere is dependent on not only aerosol loadings in the atmosphere (e.g., AOD) but also aerosol optical or radiative properties such as single-scattering albedo (SSA). What value(s) of SSA was(were) used in the numerical simulations with the SBDART radiation transfer model and how the threshold value changes single-scattering albedo (SSA)?It will be helpful if the author may provide more details about the configurations and inputs utilized in the simulations.

3. Is it necessary to use both virtual potential temperature gradient ($\frac{\partial \theta_v}{\partial z}$) and pseudoequivalent potential temperature gradient ($\frac{\partial \theta_{se}}{\partial z}$) to define the atmospheric stability since both have very similar time-height cross section distribution patterns? Please provide a description on how to use these two gradients to define the atmospheric stability and what are the advantages of using these two gradients rather than $\frac{\partial \theta}{\partial z}$ in determining the atmospheric stability?

4. Figs. 2-3: It is suggested to replot these figures by including specific months and dates in x-axis for a better view. In addition, right y-axis should be $PM_{2.5}$ rather than PM for both figures. Please correct them.

5. Fig.3a: Usually, higher $PM_{2.5}$ concentrations, lower surface-reaching shortwave radiation, and weaker turbulent activity (i.e., lower TKE). However, such a relationship is not clear in the ABL on day 1 for Episode II and day 4 for Episode III.

6. L250-251, For the statement of "the atmospheric stratification during Episodes I and II was altered" , please provide specific calculation to illustrate how the stratification was altered". Similar statements were also found in several places in the manuscript.

7. Fig.4: It is difficult to understand that aerosol radiative forcing at top of the atmospheric column (TOA) has so close relationship with surface $PM_{2.5}$ concentrations. Please provide an explanation. Again, it is better to calculate the ARF for the integrated ABL rather than the interior of the atmospheric column.

8. Why did the authors use the absolute value of difference between SFC and ATM? Why not use ATM–SFC since ATM is positive and SFC is negative? In fact, the ATM-SFC represent a combined impact of aerosol radiative effect on surface-reaching shortwave radiation and the atmospheric layer. It is not surprised to see ATM-SFC increases with increasing $PM_{2.5}$ concentrations (see Fig.4d). Here the authors still use scatter plots to quantify the relationship between aerosol radiative effect and surface $PM_{2.5}$ in terms of model results. Are there any observational data available to verify the results?

9. Fig.6: Please add a), b), c), and d) each panel, respectively, and specify clearly in the figure caption.

10. L87-91: This is definitely not true if the authors claimed that "this paper is the first time to analyze the interaction between ….". Many studies have devoted to understanding and quantifying the interactions between aerosol radiative effect and the atmospheric boundary layer thermodynamic and dynamic structures up to now. Some examples include Zhao et al., 2019, Zhang et al., 2020, Miao et al., 2020, Liu et al. 2020, etc.

11. Line 510: Again, this study is definitely not the first one. Please delete any statement like this.

12. L15: I am very concerned with the statement with "…because most studies have been superficial". Please delete or modify it.

---

## Author Response (AR1)

**General comments:** The interaction between aerosols and the boundary layer is a hot topic in the study of the formation mechanism of air pollution in polluted areas. The aim of this paper is to evaluate the fundamental interaction between PM and ABL structure and to further quantitatively estimate the effect of aerosol radiative forcing (ARF) on ABL structure. The paper addressed relevant scientific questions and presented novel concepts, ideas and tools. The scientific methods and assumptions were almost valid and clearly outlined so that substantial conclusions were reached. The description of experiments and calculations were almost complete and precise to allow their reproduction by fellow scientists except some points. I think the manuscript could be considered to be accepted after major revision.

Response: Thank the reviewer for the encouragements and constructive suggestions. According to the reviewer's suggestions, we have done our best to revise our manuscript. The modifications have been highlighted in red in the following marked-up manuscript version.

**Major comments:**

1. Seven cases spanning two months were selected in the paper to discuss the threshold value of aerosol radiative forcing's effect on the contaminated areas' boundary layer structure. Are these cases representative, and would the thresholds change in other cases?

Response: Thank the reviewer for the comments. Firstly, this campaign was launched in Beijing city to obtain the vertical profile observations of meteorological elements in the boundary layer. This experiment lasted from November 2018 to January 2019, and we obtained two-month data sets that can reflect the atmospheric boundary layer structure and atmospheric pollution in winter in Beijing. Second, we need to restate that the threshold value of aerosol radiative forcing's effect on the boundary layer structure was obtained based on the whole two-month data rather than the several cases. Only in the qualitative analysis of the relationship between the aerosol radiation effect and the boundary layer we selected cases to analyze and explain. It means the Figs. 4-7 involved in the quantitative analysis of aerosol radiative forcing influences on the boundary layer structure were processed and obtained based on the whole two-month datasets. We think the threshold value results could be representative and reflect specific effects of aerosol radiation forcing on boundary layer structure in winter in Beijing.

2. With only a finite number of points in Fig. 4, does the current fitting relationship pass the significance test?

Response: Thank the reviewer for the comments and suggestions. We need to explain that the current fitting relationships in Fig. 4 have passed the significant test. More details were shown below:

We used SPSS V19.0 software to calculate the relationship coefficients between $PM_{2.5}$

and TOA, ATM, SFC, and |SFC-ATM|, respectively, shown in Table 1. The significance levels between $PM_{2.5}$ and TOA, SFC, and |SFC-ATM| are respectively less than 0.01, indicating that they have passed the 99% significance test and have a significant correlation, respectively. The significance level between $PM_{2.5}$ and ATM is 0.021, grater than 0.01 and less than 0.05, indicating that they have passed the 95% significance test and have a reasonable correlation.

Table 1. Relationship test

| | | N | Relationship coefficient ($R^2$) | Significance level |
|---|---|---|---|---|
| a | $PM_{2.5}$ & TOA | 13 | 0.75 | 0.000 |
| b | $PM_{2.5}$ & ATM | 13 | 0.40 | 0.021 |
| c | $PM_{2.5}$ & SFC | 13 | 0.83 | 0.000 |
| d | $PM_{2.5}$ & |SFC-ATM| | 13 | 0.81 | 0.000 |

3. In Fig. 1b, the results for aerosol radiative forcing have values only for individual moments of the day, and a detailed explanation of how they relate to hourly variations in atmospheric conditions and PM concentrations is needed.

Response: Thank the reviewer for the constructive suggestions. According to the reviewer's suggestions, we have added a detailed explanation of how the aerosol radiative forcing relates to hourly variations in atmospheric conditions and PM concentrations in the revised manuscript.

4. What is the physical mechanism by which |SFC-ATM| affects the threshold of atmospheric stability?

Response: Thank the reviewer for the comments. |SFC-ATM|, defined as the absolute value of the difference between SFC and ATM, represents aerosols' combined action on the solar radiation reaching the aerosol layer and the ground. Larger values of |SFC-ATM| indicate either stronger aerosol scattering (higher SFC) or absorption effects (higher ATM), or indicate both stronger aerosol scattering and absorption effects. No matter which one causes the increased |SFC-ATM|, they all imply a more significant temperature difference between the surface and the above atmosphere layer. That means a higher |SFC-ATM| would lead to a more stable atmospheric stratification, which would suppress the turbulence development.

5. When calculating TKE, why a one-hour wind standard deviation was chosen rather than a half-hour or two-hour standard deviation? In lines 141-152, the temporal and spatial scales of TKE need to be clarified.

Response: Thank the reviewer for the comments and suggestions. Considering the time series of the boundary layer meteorological elements profile were displayed on the hourly scale, we choose to calculate one-hour TKE for better analyze the relationship among them. Regarding the temporal and spatial scales of TKE have been added in the

calculation part of Section 2.

6. Fig. 1 is of low quality and should be improved. In Fig. 1-(a)-III, why does the PM not increase with decreasing ABLH?

Response: Thank the reviewer for the comments and suggestions, and we have improved the quality of Fig.1. In Fig.1-(a)-III, $PM_{2.5}$ concentrations were generally below ~40 µg m$^{-3}$, when there was a decreasing ABLH, the $PM_{2.5}$ concentrations have slightly increased. The $PM_{2.5}$ concentrations did not increase as significantly as those in heavy pollution phases of cases I and II. Due to the drop of vertical diffusion height, the PM was accumulated at the ground level, increasing the surface PM concentrations to some degree. However, during the clean period III, Beijing was controlled by clean and dry winds, the air humidity was quite low. With less PM loading and low humidity near the surface, the heterogeneous reaction was not intense. The weak secondary aerosol formation would not lead to an outbreak of PM concentrations near the surface. That is the reason that the PM did not increase much with decreasing ABLH.

[Figure]

Figure 1. Temporal evolution of (a) the PM mass concentration and atmospheric boundary layer height (PM$_{2.5}$: solid pink lines; PM$_{10}$: solid red lines; ABLH: solid blue

lines), (b) aerosol radiative forcing at the top (TOA; green bars), surface (SFC; blue bars) and interior of the atmospheric column (ATM; red bars), and (c) horizontal wind vector profiles (shaded colors: wind speeds; white arrows: wind vectors) during the typical haze pollution episodes of I (2018/12/13-16) and II (2019/1/5-8) as well as the typical clean period of III (2018/12/27-30).

7. In Figure 1-(b) I and II, the TOA varied significantly. What is the reason?

Response: Thank the reviewer for the comments and suggestions. As shown in Fig. 4, TOA forcing was proportional to the $PM_{2.5}$ concentration. With the increase in $PM_{2.5}$ concentration, elevated aerosol loading near the surface would scatter more solar radiation back into outer space and cause less solar radiation reaching the ground, corresponding to a cooling of the surface and making negative SFC. TOA means the aerosol radiative forcing at the top of the atmosphere column and is the sum of ATM and SFC. Considering that anthropogenic aerosols are mostly scattering aerosols, the SFC forcing is generally stronger than ATM, corresponding to a cooling of the earth-atmosphere system. The TOA forcing was thus usually negative and had a similar trend with SFC. Thus, in Figure 1-(b) I and II, with PM concentrations increasing, the TOA varied significantly.

8. There are very interesting results for PM and temperature in Figure 2. What are the diurnal characteristics of the potential temperature? Does potential temperature affect the diurnal concentration of $PM_{2.5}$?

Response: Thank the reviewer for the comments and suggestions. Figure 2 shows temporal variations in the vertical profiles of (a) the virtual potential temperature gradient ($\partial\theta v/\partial z$), (b) pseudoequivalent potential temperature gradient ($\partial\theta se/\partial z$) and (c) temperature inversion phenomenon (shaded colors: inversion intensity) during the typical haze pollution episodes of I (2018/12/13-16) and II (2019/1/5-8) as well as the typical clean period of III (2018/12/27-30). Figures 2(c) have shown the relationship between PM and temperature structure. For example, when the temperature vertical gradient is positive means a temperature inversion occurs. This abnormal temperature structure would lead to a stable stratification with a positive potential temperature gradient. Figure 2(a)-(b) exactly present the potential temperature conditions corresponding to the temperature structure in Fig. 2(c). The temporal variations in the vertical profiles of (a) the virtual potential temperature gradient ($\partial\theta v/\partial z$) and (b) pseudoequivalent potential temperature gradient ($\partial\theta se/\partial z$) can represent a diurnal variation in potential temperature (stratification stability) which influence the diurnal change in $PM_{2.5}$. The specific analysis was provided in section 3.2.

9. In Figure 4, other dots represent mean data. How is it calculated?

Response: Thank the reviewer for the comments. In Fig. 4, the other dots represent mean data calculated by averaging the daily data at a fixed step length. The daily data means daily mean values of TOA, ATM, SFC, and corresponding daily averaged $PM_{2.5}$ mass concentration from 27 November 2018 to 25 January 2019 in Beijing. The mean

PM$_{2.5}$ concentrations were obtained by averaging daily PM$_{2.5}$ concentrations at intervals of 10 μg m$^{-3}$. The mean TOA, ATM, and SFC were obtained after the corresponding daily TOA, ATM, and SFC average, respectively. For example, all daily PM$_{2.5}$ concentrations greater than 40 μg m$^{-3}$ and less than 50 μg m$^{-3}$ were averaged as a mean PM$_{2.5}$ concentration, and TOA values (ATM; SFC) corresponding to this daily PM$_{2.5}$ concentration range were also averaged as a mean TOA (ATM; SFC). We have added a more detailed calculation description in the Fig. 4 caption.

10. The empirical relationships of TKE and |SFC-ATM| are very interesting in Figure 6 (left upper panel). It established the thermodynamic relationship between ARF and TKE by using the measured data. Why does the fitting relationship fit so well below 300 meters?

Response: Thank the reviewer for the comments. As we can see in Fig. 6, the exponential relationship between TKE and |SFC-ATM| was notable in the lower layers (below ~300 m) and gradually deteriorated with the increasing altitude. We all know that aerosols are mainly concentrated in the lower atmosphere, contributing the most to the SFC and ATM forcing. The stratification stability induced by the aerosol radiative effect would mainly occur in lower layers. The much better exponential relationship between TKE and |SFC-ATM| in the lower layers exactly further confirmed that the considerable change in atmospheric stratification caused by aerosols indeed existed and was mainly shown in the lower layers. With the increase of altitude, aerosol loading is in decline; thus, aerosol radiative effect on the atmospheric stability drops. Furthermore, at a relatively high altitude, the aerosol is few, and the radiation effect has almost no influence on the stability of the atmosphere layer.

11. The ARF threshold is about 55 W m$^{-2}$. What about the concentration of PM$_{2.5}$? Is it possible to derive a threshold concentration for PM$_{2.5}$ based on current observational relationships. The PM$_{2.5}$ threshold would be a very meaningful target for air pollution control.

Response: Thank the reviewer for the comments and constructive suggestions. As we can see from Fig. 4(d), the exponential relationship between PM$_{2.5}$ and |SFC-ATM| was founded. According to the linear fitting equation of y=0.49x+31.21 (x: PM$_{2.5}$; y: |SFC-ATM|), it is possible to derive a threshold concentration for PM$_{2.5}$ based on the current |SFC-ATM| threshold of about 55 W m$^{-2}$.

[Figure]

Figure 4. Scatter plots of the $PM_{2.5}$ mass concentration (x) versus aerosol radiative forcing at the surface (SFC; y; a), interior of the atmospheric column (ATM; y; b) and top of the atmospheric column (TOA; y; c) as well as the absolute difference of SFC and ATM (|SFC-ATM|; y; d), respectively (gray dots: daily data; other dots: mean data). (The daily data means daily mean values of TOA, ATM, SFC, and corresponding daily averaged $PM_{2.5}$ mass concentration from 27 November 2018 to 25 January 2019 in Beijing. The mean $PM_{2.5}$ concentrations were obtained by averaging daily $PM_{2.5}$ concentrations at intervals of 10 µg m$^{-3}$. The mean TOA, ATM, and SFC were obtained after the corresponding daily TOA, ATM, and SFC average, respectively. For example, all daily $PM_{2.5}$ concentrations greater than 40 µg m$^{-3}$ and less than 50 µg m$^{-3}$ were averaged as a mean $PM_{2.5}$ concentration, and TOA values (ATM; SFC) corresponding to this daily $PM_{2.5}$ concentration range were also averaged as a mean TOA (ATM; SFC)).

12. The review of aerosol radiative forcing in the introduction needs to be strengthened.

Response: Thank the reviewer for the comments and constructive suggestions. As the reviewer's suggested, the review of aerosol radiative forcing in the introduction has been strengthened.

13. Conclusion needs to be subdivided and further simplified.

Response: Thank the reviewer for the comments and constructive suggestions. As you suggested, the Conclusion has been subdivided and further simplified.

14. In Figure 8, TKE >2 m$^2$ s$^{-2}$, |SFC-ATM| ~55 W m$^{-2}$. Are these thresholds generalizable?

Response: Thank the reviewer for the comments. Firstly, this campaign was launched in Beijing city to obtain the vertical profile observations of meteorological elements in the boundary layer. This experiment lasted from November 2018 to January 2019, and we obtained two-month data sets that can reflect the atmospheric boundary layer structure and atmospheric pollution in winter in Beijing. Second, the threshold value of aerosol radiative forcing's effect on the boundary layer structure was obtained based on the whole two-month data. It means Fig. 8 involved in the quantitative analysis of aerosol radiative forcing influences on the boundary layer structure were processed and obtained based on the whole two-month datasets. We think the threshold value results could be representative and reflect specific effects of aerosol radiation forcing on boundary layer structure in winter in Beijing.

**Minor comments:**

English writing should be polished. Some sentences were hard to read.

1. e.g. line 18-20 "Multi-episode contrastive analysis stated the key to determining whether haze outbreak or dissipation was the ABL structure (i.e., stability and turbulence kinetic energy (TKE)) satisfied relevant conditions." Should be "Multi-period comparative analysis indicated that the key to determining whether the haze outbreak or dissipation occurs is whether the ABL structure (i.e., stability and turbulent kinetic energy (TKE)) satisfies the relevant conditions."

Response: Thank the reviewer very much for this gramma suggestion. We have corrected it.

2. Line 22-23. "SFC and ATM is respectively the ARF at the surface and interior of the atmospheric column" should be "SFC and ATM are the ARFs at the surface and interior of the atmospheric column, respectively."

Response: Thank the reviewer very much for this gramma suggestion. We have corrected it.

3. Line 37-38. (Li et al., 2020; Xu et al., 2019), should be cited at the end of this sentence.

Response: Thank the reviewer for this suggestion. We have corrected it.

4. Line 316 two "dropped to".

Response: Thank the reviewer for pointing out this mistake. We have corrected it.

**Anonymous Referee #2

1. The authors attempted to propose a parameter, $|SFC{-}ATM|$ for quantification of the impact of aerosol radiative forcing (ARF) on the atmospheric boundary layer (ABL) structure. Why did the author use the ARF of the interior of the atmosphere column (ATM) rather than the ARF in the ABL since most of aerosols or particulate matters are trapped in the atmospheric boundary layer?

Response: Thanks for the reviewer's comment. First of all, when quantifying the impact of aerosols on climate change, it is more to judge its impact on the earth-atmosphere system as a whole, so the top of the atmosphere's choice will be more reasonable. Secondly, in our previous work, we used the path radiation in MODIS data as a key parameter for calculating the atmospheric SSA, which represents the radiation value at the top of the atmosphere (TOA). In order to facilitate the comparison and verification in the later calculation process, we chose the same height to calculate the relevant radiation results, which can perform unified calculation and analysis both on the top of the atmosphere. We believe this can get more representative results in aerosol radiation research (Gong et al., 2014; Lee et al., 2007; Xin et al., 2016). Finally, as the reviewer said, aerosols are concentrated in the boundary layer and few in the stratosphere. It is because most aerosols exist in the boundary layer that we have verified in the previous sensitivity test, and the calculations at the top of the boundary layer and the top of the atmosphere are as follows:

$$\Delta F^{AEROSOL} = (\Delta F_{aero}^{TOA} - \Delta F_{non-aero}^{TOA}) - (\Delta F_{aero}^{SFC} - \Delta F_{non-aero}^{SFC}) \qquad (1)$$

$$\Delta F = F^{\text{downward}} - F^{\text{upward}} \qquad (2)$$

Where $\Delta F$ denotes the net downward flux (downward minus upward radiation); the subscripts "TOA" and "SFC" denote the top of the atmosphere/boundary layer and the surface; and "aero" and "non-aero" denote dusty and clean skies (Chou et al., 2002). Since there are few aerosols at high altitudes, the $\Delta F$ aero- $\Delta F$ non-aero itself is derived from the boundary layer difference. The $\Delta F$ aero- $\Delta F$ non-aero at high altitudes is negligible. So the radiative forcing generated by aerosols will not be significantly different because of the ABL or the top of the atmosphere. For these three reasons, we finally chose the top of the atmosphere for analysis.

Chou, M., Chan, P., and Wang, M.: Aerosol radiative forcing derived from SeaWiFS-retrieved aerosol optical properties, J. Atmos. Sci., 59, 748–757, http://dx.doi.org/10.1175/1520-0469(2002), 2002.

Gong, C., Xin, J., Wang, S., Wang, Y., Wang, P., Wang, L., and Li, P.: The aerosol direct radiative forcing over the Beijing metropolitan area from 2004 to 2011, J. Aerosol Sci., 69, 62-70, https://doi.org/10.1016/j.jaerosci.2013.12.007, 2014.

Xin, J., Gong, C., Wang, S., and Wang, Y.: Aerosol direct radiative forcing in desert and semi-desert regions of northwestern China, Atmos. Res., 171, 56-65, https://doi.org/10.1016/j.atmosres.2015.12.004, 2016.

Lee K., Li Z., Wong M., Xin J., Wang Y., Hao W., and Zhao F.: Aerosol single scattering albedo estimated across China from a combination of ground and satellite measurements, J. Geophys. Res.: Atmos., 112(D22), https://doi.org/10.1029/2007JD009077, 2007.

2. Impact of ARF on reduction of surface-reaching shortwave radiation and heating/cooling of the atmosphere is dependent on not only aerosol loadings in the atmosphere (e.g., AOD) but also aerosol optical or radiative properties such as single-scattering albedo (SSA). What value(s) of SSA was (were) used in the numerical simulations with the SBDART radiation transfer model and how the threshold value changes single-scattering albedo (SSA)? It will be helpful if the author may provide more details about the configurations and inputs utilized in the simulations.

Response: Thanks for the reviewer's comment. We have added relative details in the manuscript after the first referee round. "The algorithm of SBDART (Santa Barbara DISORT Atmospheric Radiative Transfer) (Levy et al., 2007) is the core model to calculate the radiative forcing parameters. A standard mid-latitude atmosphere is used in SBDART in Beijing. AOD and Angstrom Exponent (AE) at 550 nm were obtained from sun-photometer. Multiple sets of Single Scattering Albedo (SSA) and backscattering coefficient were calculated based on MIE theory, and surface albedo & path radiation were read from MODIS (MOD04), which is used to calculate radiative forcing at the top of atmosphere (TOA). The TOA results were combined with MODIS observations, the result which has the lowest deviation is defined as the actual parameters of aerosols, and this set of parameters would be used to calculate the radiative forcing at the surface, top, and interior of the atmospheric column (Gong et al., 2014). Hourly radiative forcing parameters, including the ARF at the top (TOA), surface (SFC), and interior of the atmospheric column (ATM) at an observation site in Beijing can be calculated based on this algorithm. More detailed descriptions are provided in our previous work (Gong et al., 2014; Xin et al., 2016)." was added in Section 2.

3. Is it necessary to use both virtual potential temperature gradient and pseudoequivalent potential temperature gradient to define the atmospheric stability since both have very similar time-height cross section distribution patterns? Please provide a description on how to use these two gradients to define the atmospheric stability and what are the advantages of using these two gradients rather than potential temperature gradient in determining the atmospheric stability?

Response: Thank the reviewer very much for this comment. Using both virtual potential temperature gradient and pseudoequivalent potential temperature gradient to define the atmospheric stability is more accurate and closer to the real atmosphere condition. Because the real atmosphere consists of saturated and unsaturated air masses. The negative virtual potential temperature gradient means absolute unstable stratifications for both saturated and unsaturated air masses, rare except in the lower layers where it is possible. When the virtual potential temperature gradient is positive while the

pseudoequivalent potential temperature gradient is negative means a stratification of conditional instability. The atmosphere stratification is unstable for a saturated air mass and stable for an unsaturated air mass. The stratification of conditional instability will become unstable once the saturated air mass reaches the condensation height due to strong local convection or substantial uplift of dynamic factors. The positive pseudoequivalent potential temperature gradient means absolute stable stratifications for both saturated and unsaturated air masses. However, the potential temperature gradient in determining the atmospheric stability only refers to unsaturated air masses. These are the reason that we choose to use both virtual potential temperature gradient and pseudoequivalent potential temperature gradient to define the atmospheric stability.

4.  Figs. 2-3: It is suggested to replot these figures by including specific months and dates in x-axis for a better view. In addition, right y-axis should be $PM_{2.5}$ rather than PM for both figures. Please correct them.

Response: Thank the reviewer very much for this comment and suggestion. As you suggested, we have replotted Fig. 1-3 to add specific months and dates in the x-axis, shown below. However, the right y-axis should be PM mass concentration for both time series of $PM_{2.5}$ and $PM_{10}$ concentration has been plotted.

[Figure]

Figure 1. Temporal evolution of (a) the PM mass concentration and atmospheric boundary layer height (PM$_{2.5}$: solid pink lines; PM$_{10}$: solid red lines; ABLH: solid blue

lines), (b) aerosol radiative forcing at the top (TOA; green bars), surface (SFC; blue bars) and interior of the atmospheric column (ATM; red bars), and (c) horizontal wind vector profiles (shaded colors: wind speeds; white arrows: wind vectors) during the typical haze pollution episodes of I (2018/12/13-16) and II (2019/1/5-8) as well as the typical clean period of III (2018/12/27-30).

[Figure]

Figure 2. Temporal variation in the vertical profiles of (a) the virtual potential temperature gradient ($\partial\theta$v/$\partial$z), (b) pseudoequivalent potential temperature gradient ($\partial\theta$se/$\partial$z) and (c) temperature inversion phenomenon (shaded colors: inversion intensity) during the typical haze pollution episodes of I (2018/12/13-16) and II (2019/1/5-8) as well as the typical clean period of III (2018/12/27-30).

[Figure]

Figure 3. Temporal variation in the vertical profiles of (a) the turbulent activity (shaded colors: TKE), (b) atmospheric humidity (shaded colors: vapor density) and (c) vertical distribution of suspended particles (shaded colors: BSC) during the typical haze

pollution episodes of I (2018/12/13-16) and II (2019/1/5-8) as well as the typical clean period of III (2018/12/27-30).

5. Fig.3a: Usually, higher $PM_{2.5}$ concentrations, lower surface-reaching shortwave radiation, and weaker turbulent activity (i.e., lower TKE). However, such a relationship is not clear in the ABL on day 1 for Episode II and day 4 for Episode III.

Response: Thank the reviewer very much for this comment. Usually, in the daytime of the clean day, with the mixing layer developing the turbulent activity would be strong. In the ABL on day 1 for Episode II and day 4 for Episode III, the $PM_{2.5}$ concentrations were really low while the turbulent activity (i.e., lower TKE) was not too strong. Both mechanical and thermal actions determine turbulence activity. The wind fields during day 1 for Episode II and day 4 for Episode III were relatively weak, while the other clean periods were always corresponding to strong winds. With little mechanical action on turbulence generation, the TKE during these periods were not as strong as other clean periods.

6. L250-251, For the statement of "the atmospheric stratification during Episodes I and II was altered", please provide specific calculation to illustrate how the stratification was altered". Similar statements were also found in several places in the manuscript.

Response: Thank the reviewer very much for this comment. Regarding the statement "the atmospheric stratification during Episodes I and II was altered" in line 250-251 was concluded based on the previous analysis. The specific description is shown below:

"During the remainder of the 2nd day, the PM mass concentration continued to increase with south winds blowing and reached its highest level at midnight with a $PM_{2.5}/PM_{10}$ mass concentration of ~110/150 μg m$^{-3}$ during both episodes I and II. The highest BSC values mainly occurred from the ground to a height of 1 km at this time, implying that a portion of the suspended particles was pushed down to the near-surface. *Noteworthily, regardless of the wind field, the atmospheric stratification states during this rising phase changed more notably. Before southerly wind transport occurred, the evolution of the stability indicator ($\partial\theta v/\partial z$; $\partial\theta se/\partial z$) profiles during episodes I and II was analogous to that during episode III (Figs. 2(a)- (b)). The stratification states at the different heights (0-1 km) were either unstable or neutral, with negative or zero $\partial\theta v/\partial z$ values, respectively, whereby no clear nor strong temperature inversion phenomenon occurred in the lower atmosphere layer (Fig. 2(c)). The corresponding ABLHs were the same (Fig. 1(a)). However, the atmospheric stratification from ~0.5-1 km during the episode I and from 0-1 km during episode II became quite stable during the PM increase period, with positive values of $\partial\theta se/\partial z$ and almost no turbulent activity (TKE: ~0 m$^2$ s$^{-2}$) (Fig. 3(a)). In contrast to an increased ABLH during clean period III, the ABLHs during episodes I-II sharply decreased.* Considering that aerosol scattering and absorbing radiation could modify the temperature stratification (Li et al., 2010; Zhong et al., 2018), the aerosol radiation effect is too weak at a low PM level to change the latter, which defines the atmospheric stability. With the elevated PM level due to southerly transport, ARF also increased, with SFC (ATM) reaching ~-40 (~20) W m$^{-2}$

and ~-75 (~30) W m$^{-2}$ during episodes I and II, respectively. Less radiation reaching the ground and more heating the atmosphere above the ground, and in comparison to clean episode III, the atmospheric stratification during episodes I and II was altered".

As described above, with the PM rising and the ARF increasing in episodes I and II, the corresponding atmospheric stratifications were altered compared to that in clean episode III and the previous no PM rising period.

7. Fig.4: It is difficult to understand that aerosol radiative forcing at top of the atmospheric column (TOA) has so close relationship with surface PM$_{2.5}$ concentrations. Please provide an explanation. Again, it is better to calculate the ARF for the integrated ABL rather than the interior of the atmospheric column.

Response: Thank the reviewer very much for this comment. As shown in Fig. 4(a), TOA forcing was proportional to the PM$_{2.5}$ concentration. With the increase in PM$_{2.5}$ concentration, elevated aerosol loading near the surface would scatter more solar radiation back into outer space and cause less solar radiation reaching the ground, corresponding to a cooling of the surface and making negative SFC. TOA means the aerosol radiative forcing at the top of the atmosphere column and is the sum of ATM and SFC. Considering that anthropogenic aerosols are mostly scattering aerosols, the SFC forcing is generally stronger than ATM, corresponding to a cooling of the earth-atmosphere system. The TOA forcing was thus usually negative and had a similar trend with SFC. The ARF calculation for the interior of the atmospheric column rather than the integrated ABL has been explained in Question 1.

8. Why did the authors use the absolute value of difference between SFC and ATM? Why not use ATM–SFC since ATM is positive and SFC is negative? In fact, the ATM-SFC represent a combined impact of aerosol radiative effect on surface-reaching shortwave radiation and the atmospheric layer. It is not surprised to see ATM-SFC increases with increasing PM$_{2.5}$ concentrations (see Fig. 4d). Here the authors still use scatter plots to quantify the relationship between aerosol radiative effect and surface PM$_{2.5}$ in terms of model results. Are there any observational data available to verify the results?

Response: Thank the reviewer very much for this comment. First of all, we all know that the difference between SFC and ATM means a combined impact of the aerosol radiative effect on surface-reaching shortwave radiation and the atmospheric layer. The reason we use the absolute value of SFC-ATM is that ATM is positive and SFC is negative; thus the SFC- ATM is always negative. The absolute value of SFC-ATM represents the same meaning as ATM-SFC. Secondly, we plotted this scatter plot (Fig. 4d) to show the relationship between the combined impact of the aerosol radiative effect on surface-reaching shortwave radiation and the atmospheric layer and PM$_{2.5}$ concentrations. It shows |SFC-ATM| increases with increasing PM$_{2.5}$ concentrations. We need to explain that the aerosol radiative forcing (ie., SFC and ATM) can be obtained only by models. Regarding the observational data verify, Zhong et al. (2018) once verified the relationship between the global radiant exposure measured at the

surface and PM$_{2.5}$ concentrations, shown as below. To further investigate the impact of elevated PM$_{2.5}$ on the loss in surface solar radiation, they calculated daytime mean PM$_{2.5}$ mass concentration, direct, diffuse, and global radiant exposure in December 2016 to 10th January 2017 in Beijing. We can see that the radiation reaching the ground decreased with the PM$_{2.5}$ concentration increasing, consistent with the relationship between SFC and PM$_{2.5}$ concentration in Fig. 4(c). However, the radiation in the atmosphere is hard to be measured yet. Thus, the aerosol radiative effect on the earth-atmosphere system is mainly based on the aerosol radiative forcing calculated by models.

[Figure]

**Fig. 2.** The correlation of daytime mean PM$_{2.5}$ mass concentration and daytime mean radiant exposure from 1st December 2016 to 10th January 2017. (a) PM$_{2.5}$ and direct radiant exposure; (b) PM$_{2.5}$ and diffuse radiant exposure; (c) PM$_{2.5}$ and global radiant exposure; (semitransparent points represents the days with high-layer moisture, and r shows the variations without semitransparent points).

Zhong J., Zhang X., Wang Y., Liu C., and Dong Y.: Heavy aerosol pollution episodes in winter Beijing enhanced by radiative cooling effects of aerosols, Atmos. Res., 59-64, 10.1016/j.atmosres.2018.03.011, 2018.

9. Fig.6: Please add a), b), c), and d) each panel, respectively, and specify clearly in the figure caption.

Response: Thank the reviewer very much for this suggestion. We have added a), b), c), and d) each panel, respectively, and specify clearly in the figure caption, shown below.

[Figure]

Figure 6. Scatter plots of the mean absolute difference of the aerosol radiative forcing at the surface and interior of the atmospheric column (|SFC-ATM|; x) versus the mean turbulence kinetic energy (TKE; y) at the different altitudes (a; b). Scatter plots of |SFC-ATM| (x) versus TKE (y) in the ABL (c) and above the ABL (d) (gray dots: hourly data; other dots: mean data). The hourly data were collected over a two-month period in Beijing from 27 November 2018 to 25 January 2019. (The hourly data means hourly mean values of |SFC-ATM| and corresponding hourly TKE. The mean |SFC-ATM| was obtained by averaging hourly |SFC-ATM| at intervals of 10 W m$^{-2}$, then the mean TKE was obtained after the average of the corresponding hourly TKE.).

10. L87-91: This is definitely not true if the authors claimed that "this paper is the first time to analyze the interaction between ….". Many studies have devoted to understanding and quantifying the interactions between aerosol radiative effect and the atmospheric boundary layer thermodynamic and dynamic structures up to now. Some examples include Zhao et al., 2019, Zhang et al., 2020, Miao et al., 2020, Liu et al. 2020, etc.

Response: Thank the reviewer very much for this suggestion. This kind of mistake has

been pointed out, and we have corrected it in the first referee round. We thank the reviewer again for pointing out this problem and have modified it.

11. Line 510: Again, this study is definitely not the first one. Please delete any statement like this.

Response: We thank the reviewer again for pointing out this problem, and we have modified it.

12. L15: I am very concerned with the statement with "…because most studies have been superficial". Please delete or modify it.

Response: Thank the reviewer very much for this suggestion. This kind of mistake has been pointed out, and we have corrected it in the first referee round. We thank the reviewer again for pointing out this problem and have modified it.

**Marked-up manuscript:**

[revised manuscript text omitted]

$$\theta_v = T(1 + 0.608q)(\frac{1000}{P})^{0.286} \qquad (1)$$

$$\theta_{se} = T(\frac{1000}{P})^{0.286}exp\left(\frac{r_sL_v}{C_{pd}T}\right) \qquad (2)$$

where $T$ is the air temperature, $q$ is the specific humidity, $p$ is the air pressure, $r_s$ is the saturation mixing ratio, $Lv$ is the latent heat of vaporization at $2.5\times10^6$ J kg$^{-1}$ and $C_{pd}$ is the specific heat of air at 1005 J kg$^{-1}$ K$^{-1}$. All the relevant parameters can be calculated from the temperature and humidity profile data obtained with the MWR, and the values of $\theta_v$ and $\theta_{se}$ at different altitudes can be then further obtained. The hourly TKE is calculated by instantaneous three wind components sampled by Doppler wind lidar every five seconds (shown as Equation (3)-(6)). The calculated TKE profile has 
[revised manuscript text omitted]
. During the time (from 9:00 to 13:00) when the ARFs can be obtained, the ARFs showed a consistent change with PM, gradually decreased to quite low levels. Conversely, the whole ABL (0-1 km) was controlled by calm/light winds during the episode I on the 3$^{rd}$ day. On account of the calm/light winds, the horizontal wind shear sharply decreased, resulting in a decline in mechanical turbulence intensity. In the absence of an existing high PM mass concentration, strong ARF would continue to cool the ground notably and heat the aerosol layer, keeping the atmospheric stratification stable and decreasing thermal turbulence intensity. As can be seen in Fig. 1(b), SFC and TOA further increased compared to yesterday, up to ~-40 W m$^{-2}$ and ~-75 W m$^{-2}$, respectively, with ATM remaining higher (~25 W m$^{-2}$). And since the high PM concentration was relatively stable from 8:00 to 14:00 when the ARFs were obtained, the elevated ARFs also kept relatively fixed values during this time. This was different from that in case II and further indicated the sensitivity between PM concentrations and

ARFs. The ABLH barely changed on the 3$^{rd}$ day and maintained a lower altitude in the forenoon of the 4$^{th}$ day. Therefore, a rather stable atmosphere extended from ~0.3-0.5 km to ~1.5 km on the 3$^{rd}$ day and from the ground to heights of ~0.3 km in the forenoon of the 4$^{th}$ day (Figs. 2(a)-(c)). The quite low TKE was highly consistent with the atmospheric stability stratification. Since the stable stratification acted as a lid at altitudes from 0.5-1.5 km, downward momentum transport would be blocked, further explaining the lower atmosphere layer's calm/light winds. In the forenoon of the 4$^{th}$ day, it is worth noting that above the stable atmospheric stratification (0-0.3 km altitude), a relatively strong horizontal wind shear occurred corresponding to a TKE of ~1-2 m$^2$ s$^{-2}$. The accumulated particles near the surface were further inhibited right below the stable atmosphere layer, as reflected by the BSC distribution. This highlights that a stable atmosphere with a weak turbulent activity was central to pushing down the pollutant layer. The same work was exerted on the water vapor as the air humidity at this time reached ~3 g m$^{-3}$ below an altitude of ~0.3 km, accompanied by intense heterogeneous hydrolysis reactions at the moist particle surface (Zhang et al., 2008), which further increased the PM mass concentration. At noon of the 4$^{th}$ day, north winds spread down to the whole ABL, which promoted the horizontal and convective dispersion of pollutants and water vapor, and the PM mass concentration, therefore, dropped to the same level as that on clean day III. With PM$_{2.5}$ sharply dropped from ~150 μg m$^{-3}$ to ~20 μg m$^{-3}$ in four hours, the aerosol radiative effect was sensitive to PM changes and gradually decreased from 10:00 to 14:00, reaching the same level as those on clean day III finally. For case II and III, the PM concentrations barely changed during the moment (from 10:00 to 14:00), the corresponding ARFs changed little neither. Qualitatively, there was a strong correlation between the PM levels and ARFs. 
[revised manuscript text omitted]

---

## Author Response (AR2)

February 9, 2021

Dear Editor:

We are submitting our revised manuscript, entitled "The impact threshold of the aerosol radiative forcing on the boundary layer structure in the pollution region" to *Atmospheric Chemistry and Physics*.

We thank the reviewer #2 for the detailed and helpful comments to improve the manuscript. Responses to the individual comments are provided below. Reviewer comments are in **bold**. Author responses are in blue plain text. Modifications to the manuscript (Tracked changes) are highlighted in red. Line numbers in the responses correspond to those in the final submitted version.

The submitted manuscript has been revised based on reviewer #2's comments.

Sincerely,

Jinyuan Xin, Professor Institute of Atmospheric Physics, Chinese Academy of Sciences Beijing, China

**Response to Reviewer #2's comments:**

I appreciate the efforts that the authors made to address all the review comments on the original submission. However, I do find limited improvement in the revised manuscript. I still have several concerns with the conclusion and the writing in the revised version. Please see below for the detail.

We thank the reviewer for the encouragements and constructive suggestions. The response to each comment is listed below.

1. As one of the major results, the authors claimed that they were able to identify a threshold value of ARF (i.e., 55 W m-2) in determining the stability of the atmospheric boundary layer (ABL) (see lines 27-29). However, such a threshold value is determined from a scatter plot (i.e., Fig.6) with limited data points that were generated by an aerosol radiation transfer model simulation (i.e., SBDART). The simulation results are dependent on not only aerosol loadings (e.g., AOD), but also aerosol optical properties (e.g., SSA) and other metrological inputs. I believe that different model configurations including types of aerosol optical property and meteorological conditions or inputs (e.g., clouds) may have different statistical relationship as presented in Fig.6. I am not sure that this threshold value (55 W m-2) has a general meaning in Beijing and other regions. For instance, the relations between the ABL structure and the ARF parameter |SFC-ATM| (Fig.6) could be very different when a dust event with high concentration occurred in this region. This concern is not well addressed in the revised version.

Thank the reviewer for the comments and constructive suggestions.

Firstly, this campaign was launched in Beijing city to obtain the vertical profile observations of meteorological elements, time series of PM concentration and corresponding aerosol optical properties in the boundary layer. This experiment lasted from November 2018 to January 2019, and the threshold value of aerosol radiative forcing's effect on the boundary layer structure was obtained based on the whole two-month data.

Second, the ARF was obtained from SBDART taking AOD etc. as input parameters. AOD means aerosol optical depth, an important parameter characterizing the aerosol optical properties and radiation properties. Cloud screening is important for the photometers to observe AOD, thus the measurements was taken under daytime cloudless conditions to remove the impact of clouds. Because the AOD measurements need to exclude the cloud effects, we calculated the aerosol radiative forcing under clear sky conditions. The atmospheric parameter profiles of the observation period were taken as metrological inputs of the SBDART and were considered to reflect local weather conditions. Therefore, ARF is the radiation forcing generated by pure aerosols, which was calculated under certain meteorological conditions during the clear sky observation period. The corresponding measured TKE is screened under the same weather conditions.

Additionally, AOD was obtained in Beijing and represented the local aerosol properties where anthropogenic aerosols dominate with few dust aerosols. Whereas no matter whether Beijing is dominated by anthropogenic aerosol or dust aerosol during the observation period, the measured AOD represents the overall local aerosol optical depth during the observation period, and the ARF obtained through SBDART with AOD as the input parameter also represents the aerosol radiative forcing in Beijing during the observation period. Hypothetically, there is mainly dust aerosols, the ATM value would be larger. Conversely, SFC is more significant. However, |SFC-ATM| represents the overall aerosol direct radiation effect, including both scattering effect and absorption effect. Therefore, different aerosol composition of Beijing during the study period would make different AOD etc. and thus different |SFC-ATM| values. The |SFC-ATM| obtained in this study was representative in the winter (Dec. 2018-Jan. 2019) in Beijing. The threshold value obtained from the relationship between ARF and measured TKE is of certain reference significance for the study of aerosol-ABL interaction in winter under urban conditions in Beijing.

However, based on the reviewer's comments, it needs to be emphasized that the threshold will certainly vary from region to region; For example, in the dust-prone area in Northwest China, the aerosol mainly behaves absorption effect, so the ATM value is large, and the |SFC-ATM| value in the observation period must be generally different from that in Beijing where scattering aerosols dominate. The specific threshold value would vary from that of Beijing.

2. If you read the manuscript closely, many sentences are still not written carefully from scientific perspective. The statement of "once |SFC-ATM| exceeded ~55 W m-2, the ABL structure would quickly stabilize" (Lines 28-29) is an example. Are you sure that the ABL became stable in this case? Based on my experience with large eddy simulations and field experiments, the ABL was still in a weakly unstable to neutral with the heavy PM pollution conditions. There are very chances that the ABL can reach the stable status during the heavy pollution events even in the nighttime. The 2nd example is, "... poor air quality due to rapid economic growth". This is not accurate. Actually, it was mainly due to rapid increase in anthropogenic emissions or a large amount of fossil fuel consumption. For instance, the rapid growth in economy did not cause any big trouble for air quality in US over the past several decades. The 3ds one is "Heavy air pollution episodes have always occurred with persistent inversions". There are too many sentences like these, which require very careful revision. I know that ACP will provide a language edit service, but I assume they are only for language polishing and I am not sure whether they are able to correct any inaccurate descriptions behind those sentences.

Thank the reviewer very much for these constructive suggestions.

Concerning the 1st comment "Once |SFC-ATM| exceeded ~55 W m-2, the ABL structure would quickly stabilize (Lines 28-29)", we would like to clarify it in two aspects:

(1) This result came from the statistical analysis of the datasets in the winter in Beijing. As shown in Figure R1, we can find an exponential relationship between ARF and TKE. With the increase of |SFC-ATM|, TKE decreases exponentially. And TKE decreased with increasing |SFC-ATM| and hardly changed when |SFC-ATM| exceeded the critical point. Considering exponential curve characteristics, we found that once the aerosol radiative effect defined by |SFC-ATM| exceeded 50-60 W m-2 (average of ~55 W m-2), the TKE sharply decreased from ~ 2 m2 s-2 to lower than ~ 1 m2 s-2, and then changed little with further increasing |SFC-ATM|.

(2) As we previously analyzed in section 3.2, haze pollution did break out with a stable atmosphere at night: For example, in episode II (Jan 7), the level of particulate matter increased due to southerly transport in the daytime; The enhanced direct aerosol radiation effect strengthens the ground cooling, which further promotes the occurrence of stable boundary layer at night and thus the outbreak of pollution. However, even in the daytime, when severe haze pollution occurred, its significant aerosol direct radiation effect changes the vertical temperature structure largely, which will also promote the stability of the boundary layer, as shown in episode I (Dec. 15 and Dec. 16) (Figure R2, marked as dashed back lines and black arrows). The high level of particulate matter existing under the stable boundary layer in the previous night arouse very strong aerosol direct radiation forcing in the daytime, facilitating the maintenance of a stable boundary layer in the daytime. In Beijing, strong emissions, southerly transport and the high PM level existing under the previous night's stable boundary layer would make the PM concentration extremely high in the daytime and thus the aerosol direct radiation forcing, which will promote the occurrence of the strong stable boundary layer in the daytime. In turn, the diffusion of particulate matter is further inhibited, further aggravating the pollution. This is why the severe haze pollution process in Beijing often lasts for two or three days. It is the interaction between aerosol and boundary layer that makes the pollution continuously intensified and difficult to dissipate, which is the central idea that we have been emphasizing in this study. Many previous studies also measured and reported it (Zhong et al., 2018, 2019; Zhao et al., 2019): As shown in Figure R3 (marked as black arrows) and Figure R4 (marked as dashed black lines), the high PM concentration lasts for several days in Beijing, which is often accompanied by continuous temperature inversion structure (stable boundary layer). Even in the daytime, the boundary layer would be stable under heavy haze pollution condition.

About the 2nd one, we reconsidered this sentence "Most areas in China, such as the North China Plain, have suffered from poor air quality due to rapid economic growth". As a developing country, China's rapid economic development is currently largely due to the rapid industry development. It is a rapid increase in anthropogenic emissions or a large amount of fossil fuel consumption that contributes to poor air quality in most China, such as North China Plain. As the reviewer suggested, we admitted that it is not

accurate enough, and have modified it in the revised version. We checked this whole manuscript to avoid the similar mistakes.

Regarding the 3rd one "Heavy air pollution episodes have always occurred with persistent inversions", we reconsidered it and understood why the reviewer pointed out it. This kind description was too subjective and we strongly agree with the reviewer's suggestion and have corrected all the kind of description.

**Figure R1 (Figure 6 in the manuscript).** Scatter plots of the mean absolute difference of the aerosol radiative forcing at the surface and interior of the atmospheric column (|SFC-ATM|; x) versus the mean turbulence kinetic energy (TKE; y) at the different altitudes (a; b). Scatter plots of |SFC-ATM| (x) versus TKE (y) in the ABL (c) and above the ABL (d) (gray dots: hourly data; other dots: mean data). The hourly data were collected over a two-month period in Beijing from 27 November 2018 to 25 January 2019. (The hourly data means hourly mean values of |SFC-ATM| and corresponding hourly TKE. The mean |SFC-ATM| was obtained by averaging hourly |SFC-ATM| at intervals of 10 W m-2, then the mean TKE was obtained after the average of the

corresponding hourly TKE.).